# Social hierarchy influences monkeys' risky decisions
Naomi Chaix-Eichel[1,2,3,10], Ayrton Guerillon[4,5,10], Sacha Bourgeois-Gironde[6,7,8], Nicolas P. Rougier[1,2,3,11], Thomas Boraud[1,9,11] & Sébastien Ballesta[4,5,11] ✉

Primates' decision-making in economic contexts follows distinctive patterns, as initially described by Prospect Theory. Social animals, such as monkeys, live in hierarchically structured groups where individual status may influence cognitive processes, including economic decisions. We leveraged a unique dataset from a semi-free ranging macaques' group, which had continuous access to gambling tasks over several years, yielding hundreds of thousands of trials and longitudinal assessments of social hierarchy. Our findings reveal a dynamic relationship between social hierarchy and decision parameters: middle-ranking individuals displayed reduced risk aversion for potential gains but not losses. Longitudinal analyses suggested that changes in social rank were followed by corresponding shifts in risk attitudes, implying that social position, rather than inherent traits, influences decision-making patterns. While sex had no significant impact, age was primarily associated with variations in loss aversion. These results underscore the flexibility and adaptive nature of primates' cognitive biases and provide key insights into how social structures influence risk behavior, with potential implications for understanding decision-making processes in other social species, including humans.

Risk-taking plays a fundamental role in various aspects of animals' economic behaviors, including foraging, reproduction, and social interactions[1–3]. According to the Prospect Theory (PT), risky decisions in human subjects are shaped by cognitive biases, such as probability distortions and asymmetrical treatment of gains and losses[4,5]. The variability of human risk attitudes has been extensively studied, and the results remain debated: some studies suggest lifelong stability, while others report context-dependent adaptability[6–8]. Longitudinal data tracking individual risk attitudes are limited, leaving the interplay between stable traits and adaptive behaviors unresolved. We hypothesize that cognitive biases such as risk aversion are not fixed, but adaptively shaped by external social parameters.

Non-human primates (NHP) offer an advantageous model for exploring the biological and socio-cognitive mechanisms underlying decision-making, as they share key cognitive traits with humans while exhibiting less cultural complexity. Cognitive biases, indeed, have been documented in NHP[9–15], with findings revealing similarities to human decision patterns, such as probability distortion and distinct risk attitudes toward gains and losses; however, these results are inconsistent across studies[16–18]. These discrepancies are due to (i) limited sampling, (ii) disparities between the

behavioral paradigms, which rarely include both gains and losses, and (iii) a lack of information about the social-demographic factors of the population studied.

Despite growing interest, the influence of sociodemographic factors, particularly social hierarchy[19], on primates' risk-taking behavior remains largely unexplored[20]. In primate societies, high-ranking individuals often experience reduced social risks and lower mortality rates[21–23] a pattern also relevant to human societies[24,25]. Recent findings suggest that middle-ranking individuals may encounter a less predictable, more demanding, social environment, potentially affecting their physiological and psychological states[26,27]. In addition, experimental changes in monkeys' social status influenced personality traits such as "boldness"[28]. Altogether, these studies suggest that monkeys' social hierarchy influences their attitude toward risk. However, to the best of our knowledge, this hypothesis has not been assessed using cognitive tasks.

To address this issue, we examined the relationship between social hierarchy and economic decision-making in a semi-free-ranging group of Tonkean macaques (*Macaca tonkeana*). We utilized Machines for Automated Learning and Testing (MALT) to track thousands of decisions over

[1]Univ. Bordeaux, CNRS, IMN, UMR 5293, Bordeaux, France. [2]Univ. Bordeaux, CNRS, Bordeaux INP, LaBRI, UMR 5800, Talence, France. [3]Centre Inria de l'Université de Bordeaux, Talence, France. [4]Laboratoire de Neurosciences Cognitives et Adaptatives, LNCA, UMR 7364, Strasbourg, France. [5]Centre de Primatologie de l'Université de Strasbourg, Niederhausbergen, France. [6]Centre de Recherche en Economie et Droit, Université Panthéon-Assas, Paris, France. [7]Institut Jean Nicod, Département d'Etudes Cognitives, ENS, EHESS, PSL Research University, Paris, France. [8]Faculty of Law, University of Haïfa, Haïfa, Israel. [9]CHU de Bordeaux, Service de Neurologie, Bordeaux, France. [10]These authors contributed equally: Naomi Chaix-Eichel, Ayrton Guerillon. [11]These authors jointly supervised this work: Nicolas P. Rougier, Thomas Boraud, Sébastien Ballesta. ✉e-mail: ballesta@unistra.fr

https://doi.org/10.1038/s42003-026-09817-2                                                          **Article**

several years[29], enabling daily updates of dominance hierarchies[30,31]. Our task captured both gains and losses[13], providing novel insights into the dynamic impact of social hierarchy on risk attitudes, suggesting its adaptive role.

## Results
### Asymmetry of treatments between gains and losses
Adopting the PT framework, we analyzed the decisions of 18 Tonkean macaques across 1,380,190 trials over a three-year period from February 2020 to June 2023, allowing us to study risk attitudes longitudinally (Fig. 1), using chunks of 1500 decisions (Supplementary Fig. 1, see "Methods"), while simultaneously tracking Elo-rating dynamics (Fig. 2). On average, we

observe that the monkeys' risk-taking behavior aligns with PT principles (Supplementary Figs. 2 and 3). Risk attitude (parameters $\rho+$ for gains and $\rho-$ for losses, see "Methods", Fig. 3A, B, Supplementary Fig. S2), exhibited significant distinct patterns across gain and loss domains (Wilcoxon signed-rank test, $P < 0.001$). On average, $\rho+$ was positive ($\rho+ = 0.182 \pm 0.3$), while $\rho-$ was negative ($\rho- = -0.508 \pm 0.121$), indicating that individuals were risk-averse in the gain domain and risk-seeking in the loss domain. In addition, we examined the variability of PT parameters across all individuals for all chunks of 1500 decisions (Fig. 3 and Supplementary Fig. 4). We observed that the variability within individuals of $\rho$ was significantly greater in gains than in losses (Wilcoxon signed-rank test, $P < 0.001$). Overall, probabilities distortion (parameters $\alpha+$ for gains and $\alpha-$ for losses, Fig. 3D,

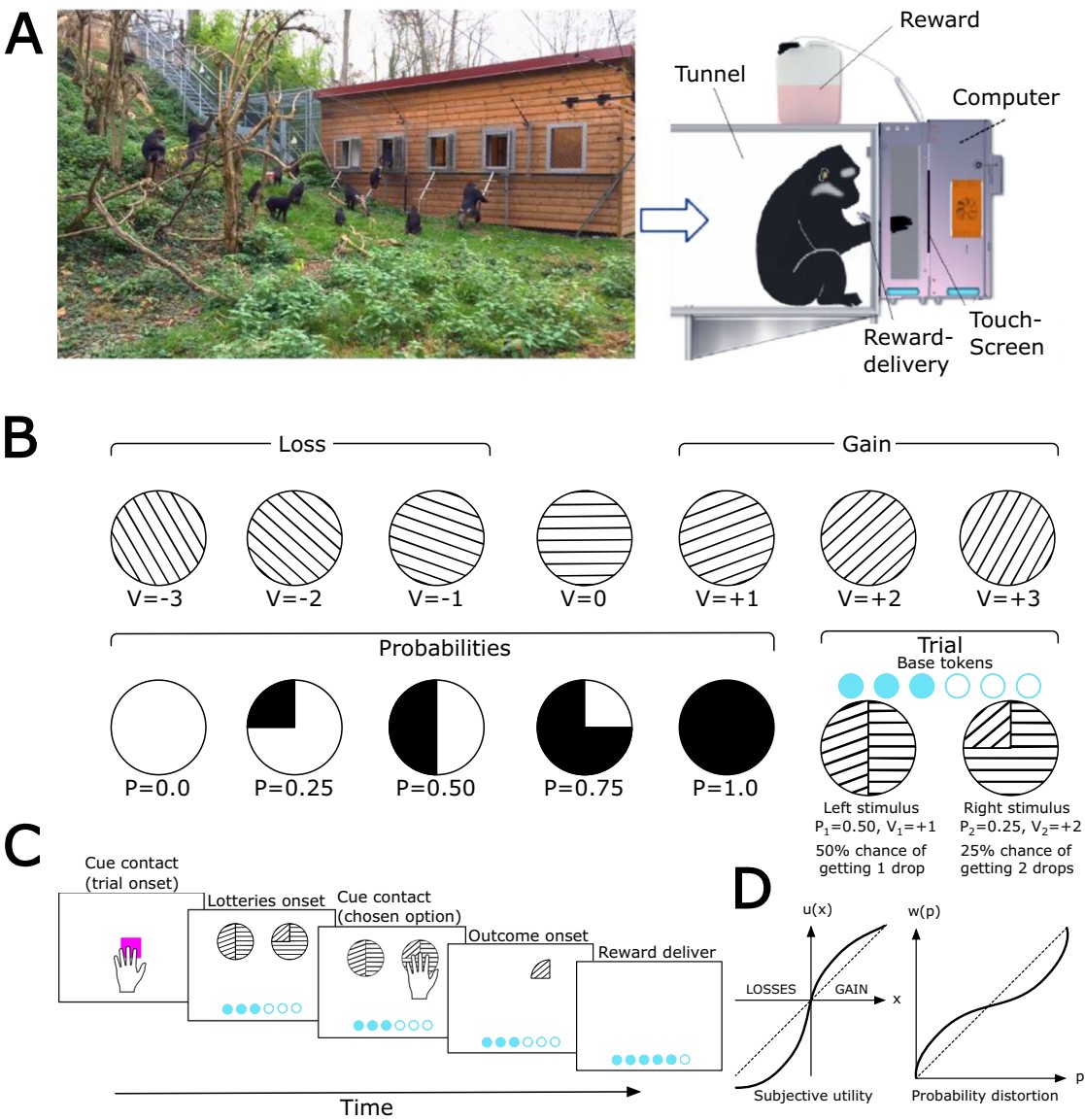

**Fig. 1 | Monkey experiment. A** Tonkean macaque's wooded park in semi-free-ranging conditions. Cognitive tasks are presented via a touchscreen interface MALT (Machine for Automated Learning and Testing). The picture was taken by Hélène Meunier from our research team. **B** Economic task. The orientation of the lines indicates a quantity. Each lottery is represented by a pie chart composed of two slices. The arc length of each slice represents the probability of the corresponding outcome (p or 1−p). Tokens on the screen represent the quantities gained by the monkey. The monkey starts each trial with three tokens displayed on the screens. Two different pie charts with the same expected value. If the monkey chooses the left pie chart, it will have a 50% chance of getting 1 token. If it chooses the right pie chart, it will have a

25% chance to get 2 tokens. While the right pie chart gives a better outcome, it is a riskier choice than the left one. A tradeoff has to be made between risk and quantity. **C** Trial example showing the order sequence to make economic decisions and access the reward. **D** Prospect Theory according to Kahneman and Tversky[4]. Left: Subjective utility function. The concavity of the curve for gains indicates risk aversion ($\rho > 0$), while the convexity of the curve for losses indicates risk seeking ($\rho < 0$). Loss aversion is indicated by a steeper curve for losses than for gains ($\lambda > 1$). Right: Probability weighting function: small probabilities are overestimated while high probabilities are underestimated ($\alpha < 1$).

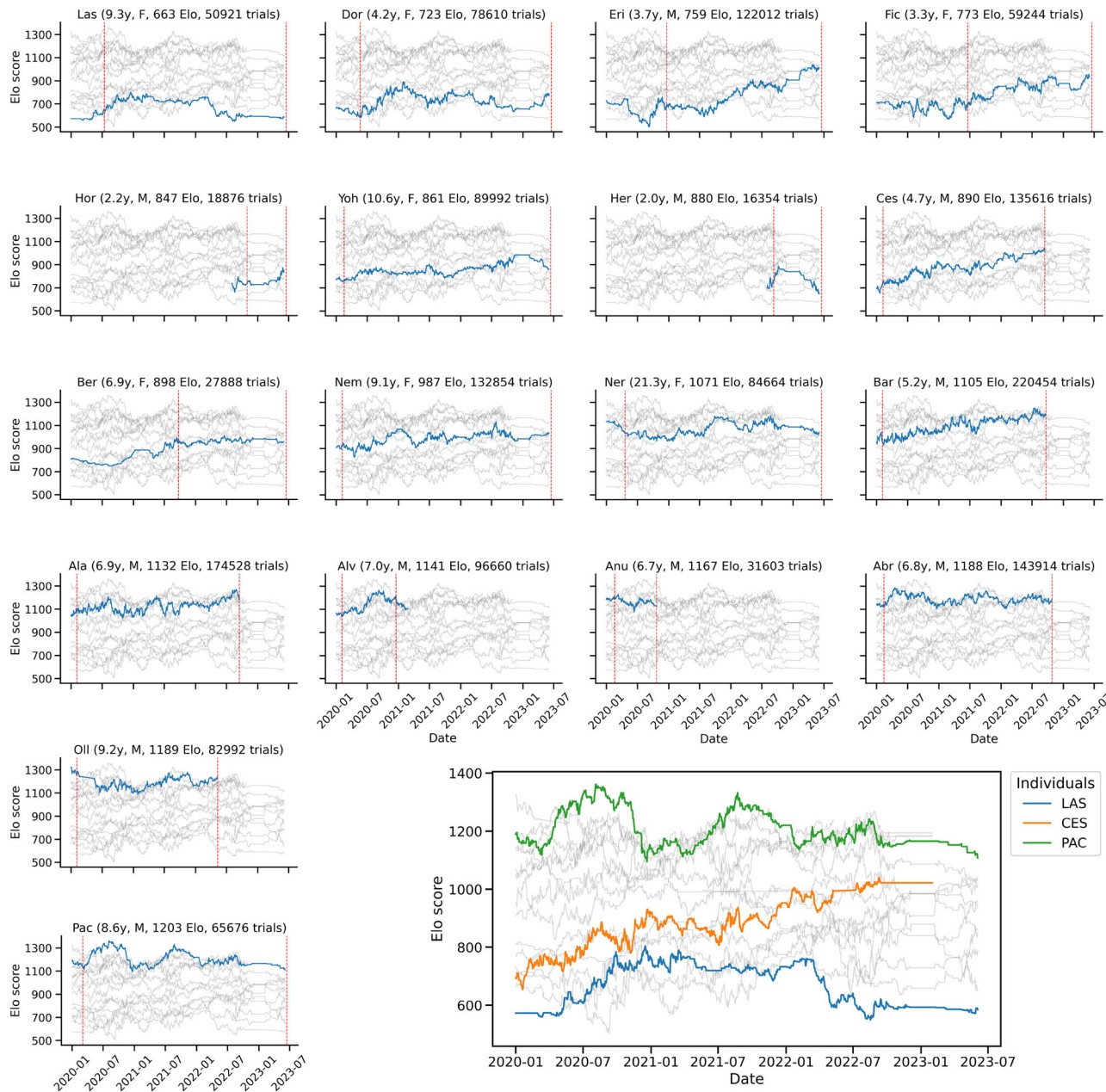

**Fig. 2 | Evolution of Elo-rating for all individuals.** Each panel's title shows the individual's name, age, sex, mean Elo-rating and total number of trials. Blue lines indicate the daily Elo-rating of the individual, while gray lines show those of others. Red dashed lines mark the first and last economic trials (training phases excluded).

The larger panel (bottom right) highlights three individuals with distinct trajectories: Pac and Las display fluctuations with increases and decreases in Elo-ratings, whereas Ces shows a steady upward trend reflecting his rise in hierarchy.

E) in gains was over 1, indicating that individuals have an S shape distortion probabilities function in gains, consistent with an underestimation of low probabilities and overestimation of high probabilities ($\alpha+ = 1.208 \pm 0.609$, Supplementary Fig. 3). In losses, $\alpha-$ was negative, indicating an inverted S shape distortion function, with an overestimation of low probabilities and underestimation of high probabilities ($\alpha- = 0.655 \pm 0.177$, Supplementary Fig. 3). Finally, loss aversion represented by $\lambda$, was consistently greater than 1, indicating that individuals were generally loss-averse ($\lambda = 2.987 \pm 1.399$, Fig. 3C and Supplementary Fig. 2).

### Dynamic dominance hierarchy and variability in conflict outcome predictability

Social rank was assessed by computing an Elo-rating based on the outcomes of dyadic conflicts for each individual's access to the MALT resource on a

daily basis, longitudinally (Figs. 1A and 2), over the whole period where they performed the economic task[30]. Elo-ratings throughout the experiments revealed a clear dominance hierarchy among individuals (Fig. 3F). For example, Lassa ("las") consistently occupied the lowest position in the hierarchy, while Patchouli ("pac") held the top rank. Importantly, this representation also illustrates the temporal variability in Elo-ratings, indicating that dominance hierarchy is rather dynamic over time (Fig. 2). Beyond ranking individuals, this method also allowed us to assess conflict outcome predictability (COP, Fig. 4F). A score of 1 indicates a clear, unambiguous predictability of conflict outcome (i.e., 100% victories or 100% defeats after dyadic conflict), while a score of 0 reflects maximum unpredictability in dominance relationships (i.e., a 50/50 split between wins and losses). When COP was correlated to Elo-ratings, a U-shaped relationship emerged (Linear Model Mixed, LMM test, $P < 0.001$). Low-ranking and

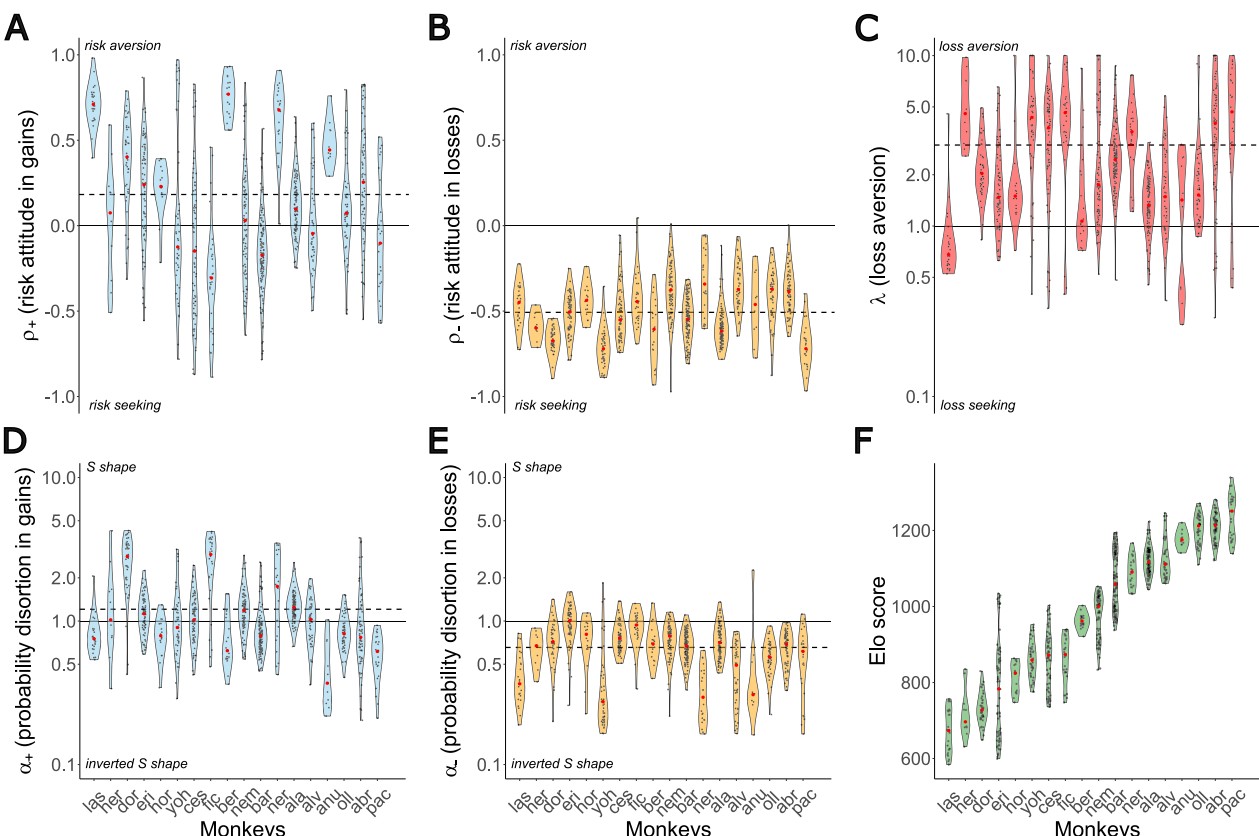

**Fig. 3 | Individual variations in social and decision-making parameters.**
**A**, **B** Violin plots displaying the variability of ρ in gains (left) and losses (right) for all individuals. The red points represent medians, while each black point corresponds to a fitted ρ parameter from 1500 trial periods per individual, separately for gain and loss trials. The dashed line indicates the mean ρ across all individuals (ρ+ = 0.182, ρ− = −0.508). The solid line represents risk neutrality (ρ = 0): values above this line (ρ > 0) indicate risk aversion, whereas values below it (ρ < 0) reflect risk-seeking behavior. **C**, **D** Violin plots showing the variability of α in gains (left) and losses (right) for all individuals. The red points denote medians. The dashed line represents the mean α across individual means (α+ = 1.208, α− = 0.655). The solid line marks no probability distortion (α = 1): values above this line (α > 1) indicate an S-shaped probability weighting function characterized by an underestimation of low probabilities and an overestimation of high probabilities. Conversely, values below

(α < 1) indicate an inverted S-shape, where low probabilities are overestimated, and high probabilities are underestimated. **E** Violin plot illustrating the variability of λ across individuals. Each black point corresponds to a fitted λ parameter from 1500 trial periods per individual. The dashed line represents the mean λ across all individuals (λ = 2.987), reflecting the loss aversion degree. The solid line represents loss neutrality (λ = 1): values above this line (λ > 1) indicate loss aversion. A logarithmic transformation has been applied to the α and λ axes to normalize the representation of behavioral variations, as changes in values between 0.1–1 and 1–10 reflect opposite but equivalent intensities of behavior. **F** Violin plots illustrating the variability of Elo-rating across individuals. In all panels, red points represent medians, while each black point corresponds to an individual's mean Elo-rating over 1500 trial periods.

high-ranking individuals exhibited high COP scores, indicating that their future conflicts tended to have more predictable outcomes. In contrast, middle-ranking individuals showed lower COP, reflecting greater unpredictability in the resolution of their future social conflicts. In summary, social hierarchy encompasses two related but distinct patterns: a linear ranking of individuals in the social hierarchy, and a U-shaped relationship reflecting the conflict outcome probabilities, where low and high-ranking monkeys have more predictable results, and mid-ranking individuals face more balanced, uncertain competitions. This non-linear effect can account for the effects of predictability of social encounters and, as described by McCowan et al., dominance certainty that represents the certainty of an individual's social status[32].

## Social hierarchy influences risk attitude for gains
To dynamically assess the effect of changes in social hierarchy on economic decision-making, we segmented each monkey's trial sequence into consecutive periods of 1500 decisions. This window size corresponds to the estimated number of trials required to reliably compute PT parameters (Supplementary Fig. 5). This segmentation enabled us to track how PT parameters evolved across successive periods. Elo-rating allows us to

compare social risk with PT parameters across the same 1500 trial periods for each individual. For statistical analyses, we used LMM to disentangle the effects of age, sex, social rank, and experience (mean trials number) of individuals while taking into account the nested nature of our data. We identify a U-shaped relationship between Elo-rating and ρ parameters for gains with ρ above 0 for low and high-ranked individuals and ρ below 0 for middle-ranking individuals (LMM test, P < 0.001, Fig. 4A, Supplementary Fig. 2, Supplementary Fig. 6A, and Table 1). These results indicate that middle-ranking monkeys are less risk-averse than those at the extremes of the hierarchy and even display risk-seeking attitudes. No significant correlation between Elo-rating and PT parameters was found in the loss domain, while all individuals displayed a comparable level of risk-seeking (LMM test, P > 0.01, Fig. 4B, Supplementary Figs. 2 and 6A, and Table 1). As observed for ρ, α parameter follows the same dynamic asymmetry between gains and losses with an inverted U-shape (LMM test, P < 0.001, Fig. 4D, E, Supplementary Figs. 3 and 6B, and Table 1). In the gain domain, individuals at the extremes of the hierarchy have an inverted S-shape probability distortion function indicating overestimation of low probabilities and underestimation of high probabilities. Middle-ranking individuals show a more neutral or S-shaped probability distortion with

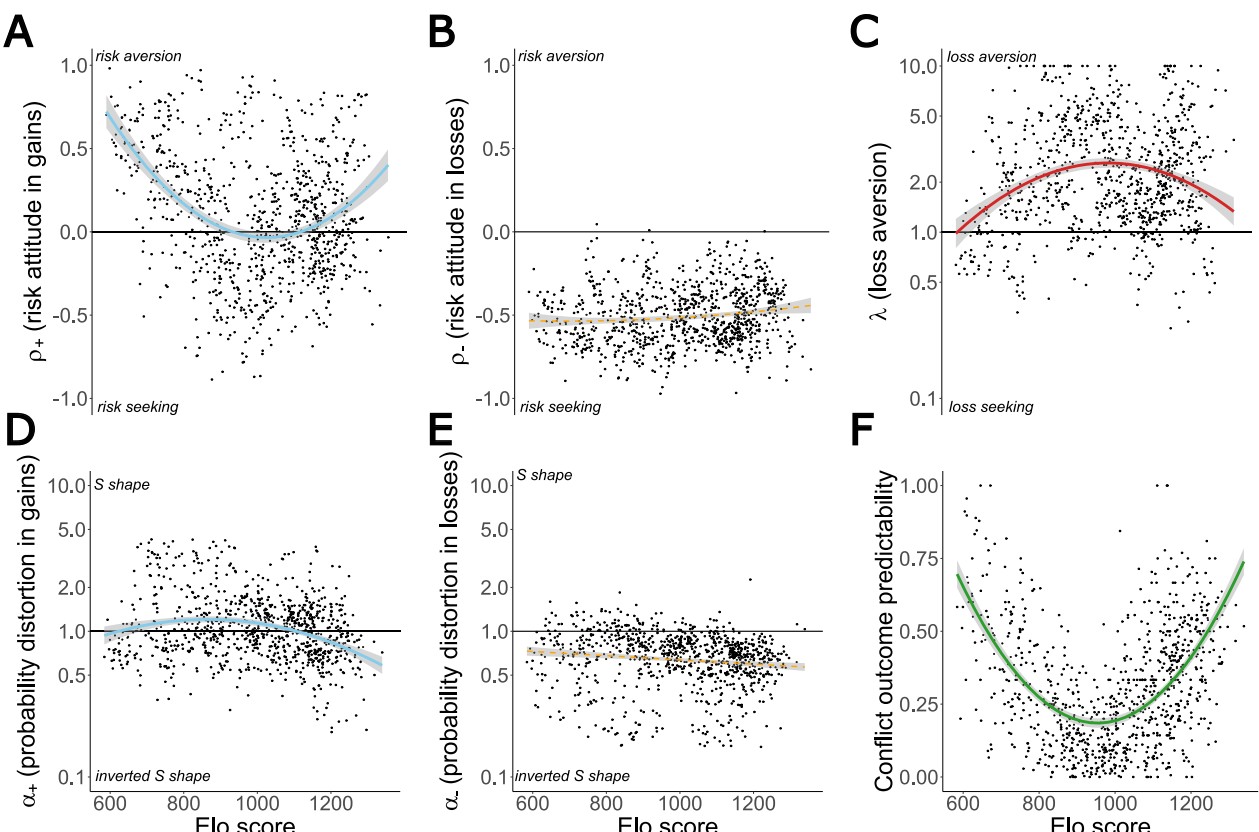

**Fig. 4 | Influence of social hierarchy on social and decision-making parameters. In all panels, each black point corresponds to a fitted social or decision-making parameter from 1500 trial periods per individual.** These 1500-trial sequences correspond to the estimated number of trials required to reliably estimate PT parameters (Supplementary Fig. 5). This segmentation enabled us to examine how PT parameters evolved across consecutive periods. **A**, **B** Relationship between ρ parameter and Elo-rating in gains (left) and losses (right). The blue curve represents a statistically significant quadratic fit ($n = 927$, $\beta = 0.08$, $P < 0.001$), highlighting a U-shaped relationship in gains while the orange dashed line represents non-significant relationship in losses ($n = 927$, $\beta = 0.01$, $P = 0.026$). The solid line represents risk neutrality ($\rho = 0$): values above this line ($\rho > 0$) indicate risk aversion, whereas values below it ($\rho < 0$) reflects risk-seeking behavior. **C** Relationship between $\lambda$ and Elo-rating. The red curve represents a statistically significant quadratic fit ($n = 927$, $\beta = -0.24$, $P < 0.001$), highlighting an inverted U-shape relationship. The solid line represents loss neutrality ($\lambda = 1$): values above this line ($\lambda > 1$)

indicate loss aversion. A logarithmic transformation has been applied to the α and λ axes to normalize the representation of behavioral variations, as changes in values between 0.1–1 and 1–10 reflect opposite but equivalent intensities of behavior. **D**, **E** Relationship between α parameter and Elo-rating in gains (left) and losses (right). The blue curve represents a statistically significant quadratic fit ($n = 927$, $\beta = -0.11$, $P < 0.001$), highlighting an inverted U-shape relationship in gains, while the orange dashed line represents a non-significant decreasing linear trend in losses ($n = 927$, $P = 0.846$). The solid line marks no probability distortion ($\alpha = 1$): values above this line ($\alpha > 1$) indicate an S-shaped probability weighting function. **F** Relationship between the conflict outcome predictability (COP) and Elo-rating. COP is computed by taking the absolute difference between the number of winning and losing conflicts during a conflict by the total number of conflicts. The green curve represents a statistically significant quadratic fit ($P < 0.001$), highlighting a U-shaped relationship.

underestimation of low and overestimation of high probabilities. In the loss domain, individuals show a constant inverted S-shape relationship. These results hold true if individuals' ordinal rank was used in the LLM instead of Elo rating (cardinal rank, Supplementary Table 2). Finally, we can observe a significant inverted U-shape relationship between $\lambda$ and Elo-rating, indicating that middle-ranking individuals have a stronger loss aversion than those at extreme hierarchy places (LMM test, $P < 0.001$, Fig. 4C, Supplementary Fig. 6B, and Table 1). Graphical representations of individual PT parameters and effects reported in LMM are provided in Supplementary Figs. 6 and 7 and individual fitting of the effect of Elo-rating for all parameters is provided in Supplementary Fig. 4. We also conducted a similar analysis in which the quadratic term of the Elo rating was replaced in the model by our measure of COP or dominance certainty (Supplementary Table 3). Our LMM model highlights that COP, which reflects the predictability of an individual's success in winning, accounts for the effect of the Elo score more strongly than dominance certainty (Supplementary Fig. 7 and Supplementary Table 3), potentially due to the chunked nature of our data.

## Influence of experience and other sociodemographic parameters

Sex does not influence risk attitude, probability distortion, or loss aversion (LMM test, $P > 0.01$, Supplementary Fig. 8 and Table 1). Age affects risk attitudes in the loss domain (Supplementary Fig. 9). Juveniles (individuals under 4 years old) exhibit a higher $\rho-$ parameter: juveniles are less risk-seeking than adults when facing losses (LMM test, $P = 0.001$, Supplementary Fig. 9B and Table 1). Moreover, loss aversion is less pronounced in juveniles (LMM test, $P = 0.005$, Supplementary Fig. 9C and Table 1) than in adults. Age also influences probability distortion in gains. Specifically, juveniles have lower α values in gains (LMM test, $P < 0.001$, Supplementary Fig. 9D and Table 1). Finally, the number of trials performed, considered as a proxy of experience in the economic task, significantly impacts risk attitudes. It is inversely correlated with $\rho+$ values, indicating that more experienced monkeys exhibit less risk aversion in gains (LMM test, $P = 0.001$, Supplementary Fig. 6D and Table 1). It is also positively correlated with higher $\rho-$ values, suggesting that experience reduces risk-seeking behavior in losses (LMM test, $P < 0.001$, Supplementary Fig. 6D and

**Table 1 | LMM models results for PT parameters**

| Predictors | $\rho+$ | | | $\rho-$ | | | $\alpha+$ | | | $\alpha-$ | | | $\lambda$ | | |
|---|---|---|---|---|---|---|---|---|---|---|---|---|---|---|---|
| | Estimates | CI | P | Estimates | CI | P | Estimates | CI | P | Estimates | CI | P | Estimates | CI | P |
| (Intercept) | −0.04 | −0.21 to 0.13 | 0.645 | −0.54 | −0.64 to −0.45 | **<0.001** | 0.09 | −0.23 to 0.42 | 0.577 | −0.54 | −0.77 to −0.32 | **<0.001** | 1.40 | 1.00 to 1.81 | **<0.001** |
| Age category [juvenile] | −0.05 | −0.25 to 0.15 | 0.631 | 0.18 | 0.07–0.29 | **0.001** | −0.79 | −1.12 to −0.45 | **<0.001** | −0.07 | −0.34 to 0.21 | 0.641 | −0.68 | −1.15 to −0.21 | **0.005** |
| Age category [subadult] | −0.07 | −0.14 to 0.00 | 0.061 | 0.01 | −0.03 to 0.05 | 0.721 | 0.11 | −0.01 to 0.23 | 0.081 | 0.07 | −0.03 to 0.18 | 0.163 | −0.22 | −0.40 to −0.05 | 0.011 |
| Sex [female] | 0.20 | −0.06 to 0.47 | 0.131 | −0.01 | −0.16 to 0.15 | 0.934 | 0.26 | −0.25 to 0.76 | 0.317 | −0.12 | −0.46 to 0.22 | 0.477 | −0.21 | −0.83 to 0.41 | 0.501 |
| Elo | −0.00 | −0.06 to 0.05 | 0.869 | −0.03 | −0.06 to −0.00 | 0.029 | −0.01 | −0.09 to 0.07 | 0.841 | 0.03 | −0.04 to 0.10 | 0.410 | −0.17 | −0.28 to −0.05 | **0.005** |
| Elo² | 0.08 | 0.05–0.10 | **<0.001** | 0.02 | 0.00–0.03 | 0.024 | −0.10 | −0.14 to −0.06 | **<0.001** | 0.03 | −0.00 to 0.06 | 0.092 | −0.25 | −0.31 to −0.20 | **<0.001** |
| Mean trial number | −0.20 | −0.22 to −0.17 | **<0.001** | 0.02 | 0.01–0.03 | **0.001** | 0.05 | 0.02–0.09 | **0.005** | −0.01 | −0.04 to 0.02 | 0.406 | 0.25 | 0.20–0.30 | **<0.001** |
| **Random effects** | | | | | | | | | | | | | | | |
| $\sigma^2$ | 0.06 | | | 0.02 | | | 0.15 | | | 0.11 | | | 0.31 | | |
| $\tau_{00}$ | 0.07 | | | 0.02 | | | 0.27 | | | 0.12 | | | 0.39 | | |
| ICC | 0.56 | | | 0.59 | | | 0.64 | | | 0.51 | | | 0.56 | | |
| N | 18 | | | 18 | | | 18 | | | 18 | | | 18 | | |
| Observations | 927 | | | 927 | | | 927 | | | 927 | | | 927 | | |
| Marginal R²/conditional R² | 0.351/0.713 | | | 0.065/0.616 | | | 0.133/0.686 | | | 0.029/0.525 | | | 0.196/0.646 | | |

All models were computed on all 1500 trial periods for all individuals after filtering the data. Values in bold represent significant effects ($P < 0.01$).

Table 1). The number of trials also impacts loss aversion (LMM test, $P < 0.001$, Supplementary Fig. 6F and Table 1), but no effect was found in probability distortion (LMM test, $P > 0.01$, Supplementary Fig. 6E and Table 1).

## Discussion

Our study confirms that monkey economic behavior, while adaptable, aligns with the principles of PT. We reveal a clear relationship between social hierarchy, experience, and risk attitude, contributing to the observed intra-individual variability in risk attitude in the gain domain and loss aversion. These findings parallel human research, where sociodemographic factors such as income, education, and employment status influence risk attitudes[33,34]. However, these relationships are often non-linear and context-dependent, shaped by interactions between individual traits, social environment, and life circumstances[35].

Our results related to the influence of social hierarchy on economic decisions highlight a consistent pattern: middle-ranking macaques exhibit reduced risk aversion in the gain domain, but this effect does not extend to the loss domain. This asymmetry underscores the differential cognitive processing of potential gains and losses, a hallmark of PT. By leveraging the dynamic nature of macaque social hierarchies, we observed that changes in an individual's social status corresponded with shifts in risk attitudes, suggesting that social position modulates the individual's physiological and mental state. This result validates our original hypothesis by supporting the view that cognitive biases such as risk aversion are not fixed, but adaptively shaped by external social pressures. NHP social hierarchies can be characterized by a linear dominance ranking that reflects an individual's relative status and a non-linear component that may be related to the predictability of social conflict outcomes. Our findings point toward the latter as a potential driver of the observed changes in risk attitudes, aligning with previous research suggesting that middle-ranking individuals experience greater social unpredictability than those at either end of the hierarchy[27]. Middle-ranking individuals are more likely to encounter peers with comparable Elo-ratings, leading to more frequent competitive interactions that may, in turn, induce behavioral[36] and physiological changes[37,38] that influence cognitive processes involved in risk-taking[39–41]. Interestingly, middle-ranking monkeys' propensity for increased risk-taking may reflect a socially adaptive strategy to navigate their ambiguous status. Competitive social contexts are known to enhance risk-seeking behavior in apes[42], and neuroendocrine responses associated with status-seeking behaviors might similarly contribute to this phenomenon in macaques[38,43]. Such adaptive flexibility could serve as a mechanism for individuals to improve their social standing while mitigating the risks of social defeat. In addition, comparing COP and dominance certainty shows that the predictability of conflict outcome is more likely to have an impact on risk attitude than dominance certainty.

Regarding sex, age, and experience, sex differences in risk-taking in humans are well-documented, with women generally being more risk-averse than men in the gain domain[44,45] but not in the loss domain[34]. Most studies attribute these differences to cultural and social factors rather than innate biological differences[46]. Research in NHP has yielded mixed results, with some studies detecting sex-based differences while others find none, often due to limited sample sizes[20]. In our study, we observed no significant sex differences across any PT parameters, suggesting that sex-based variations in risk attitudes may be more culturally driven in humans than biologically determined across primate species (but see ref. [47]). The impact of age on risk-taking behavior in humans remains inconsistent, with some studies reporting a U-shaped relationship[48,49], others noting age-related decreases or increases in risk-seeking behavior[50], and some finding no clear pattern[51]. Our data indicate that age significantly affects loss aversion in macaques, although the limited number of juvenile participants' warrants caution when interpreting this result. The potential influence of developmental factors on decision-making remains open for future investigation. Finally, the number of trials completed by an individual significantly influences their risk behavior when making decisions involving gains. The more experience the individual accumulates with the task, the less risk-averse they become. Conversely, in the loss domain, increased task exposure leads to a reduction in risk-taking behavior, although this effect is less pronounced than in the gain domain. This pattern suggests a greater flexibility in decision-making processes related to gains, as opposed to the more stable or rigid patterns observed in losses, highlighting potential differences in the underlying cognitive and affective mechanisms involved in processing gains versus losses.

Overall, our findings underscore that these risk attitudes are not merely passive reflections of individual energy budgets or resource access. The increased risk-taking behavior observed in middle-ranking individuals likely results from the uncertainty inherent in their social environment. Our results provide compelling evidence that NHP' risk attitudes are dynamically influenced by their social context, particularly through the lens of competition. This flexibility, predominantly observed in decisions involving potential gains, may reflect an adaptive strategy for navigating the competitive pressures associated with middle-ranking positions. The fact that this effect does not extend to the loss domain reinforces the asymmetrical treatment of gains and losses in primates' economic decisions. These findings emphasize that risk attitudes are not fixed traits but instead adapt to an individual's experience and the shifting social landscape of its environment. Our findings suggest that social position and experience dynamically modulate cognitive biases rather than solely reflecting inherent individual traits. These results provide novel insights into the adaptive function of primates' economic decision-making flexibility with potential implications for understanding how social structures influence risky-taking behavior across social species, including humans.

## Methods
### Subjects and ethics

Data were collected in one group of Tonkean macaques (*Macaca tonkeana*), all captive-born and housed at the Primate Centre of the University of Strasbourg, France. Animals lived in semi-free ranging conditions in a wooded park of 3788 m² with permanent access to an indoor-outdoor shelter ($2.5 \times 7.5$ m $-2 \times 4$ m). The group included 28 individuals, data from 24 individuals (13 males) were considered over a three-year period from February 2020 to June 2023. Based on outlier filtering (see below), only 18 subjects were included in the analysis. The remaining 18 Subjects (11 males) were aged $7.24 \pm 4.56$ years at the start of the experiment. All the monkeys had permanent access to water, an indoor-outdoor shelter, and enrichments. Monkeys were fed with commercial primate pellets twice a day inside the indoor shelter and received fresh fruit and vegetables once a week. Data collection was conducted non-invasively and approved by the ethical committee of the Primate Centre of the University of Strasbourg—Silabe which is authorized to house non-human primates (registration n°B6732636). The research further complied with the EU Directive 2010/63/EU for animal experiments.

### Machine for automated learning and testing (MALT)

Automated data were collected using up to four MALT, which the monkeys could access directly from their living environment in order to willingly perform several cognitive tasks[13,29]. A wooden shelter housing four MALT devices was built along the park. The four devices are spaced 65,5 cm from each other and placed 85 cm above the floor. Animals have free access to these devices through meshing tunnels (h70 × l51 × L115 cm) linking their outdoor enclosure to the wooden shelter, from the 15th of February 2016. Each MALT was accessible freely 24/7, except for 2-h cleaning and refill sessions, at least twice a week. The four MALT were placed in the same room, but were visually separated from each other by opaque Trespa® boards. Monkeys were rewarded at the device for a correct answer by receiving a sip of liquid reward (500–3000 ms of juice reward delivered by a peristaltic pump, corresponding to 0.25–1.5 mL ± 5% of diluted syrup, 1/10). Four different flavors of syrups (same sucrose concentration) were available during data collection

(exotic fruits, apple, peach, and strawberry), all MALT were filled with the same syrup after every refill session. MALT allows automatic identification of each subject due to a RFID dual-detection system[52]. For that purpose, subjects were all equipped with two RFID microchips (UNO MICRO ID/12, ISO Transponder $2.12 \times 12$ mm), injected into each forearm during the annual veterinary health check under appropriate anesthesia. When the MALT detects the RFID chip of an animal, it resumes his/her personal experimental sessions, which remain open for 30 s after the last screen touch or RFID detection.

### Hierarchy of dominance

Ballesta et al. introduced a method to measure hierarchy of dominance in the NHP community using the MALT device[30]. If an animal tries to engage with the cognitive tasks while another individual's session is active, a displacement event (mentioning a "winner" that managed to displace the "loser") is recorded in our database. Social hierarchy of the group can be computed indifferently using these displacement events or more classical observation of spontaneous social conflicts. These displacements indicate social hierarchy and dominance relationships, providing a means to assess the evolution of individual social status within the group[30,31]. Using MALT enables automatic measurement of hierarchy, with Elo-rating serving as a score to assess and track its evolution over time. Ordinal rank does not fully capture changes in social hierarchy as it provides only a snapshot of an individual's position at a given time. Conversely, Elo rating continuously updates based on each interaction, better reflecting the evolving social hierarchy and not only ranks shifts. Elo-rating is computed using the package "Elorating" in R[53]. It starts at 1000, and individuals could get or lose points depending on whether they win or lose a dyadic conflict to access the MALT and the current Elo-rating of the other individual. This method also allows us to compute a Conflict Outcome Predictability (COP) score, defined as the absolute difference between the number of wins and losses in a conflict, divided by the total number of conflicts. A dominance certainty score was assessed following the method described by McCowan et al.[27], which quantifies the uncertainty in individual dominance.

### Task and estimation of the prospect theory (PT) parameters

The gambling task used is the same as the one used in Nioche et al.[12,13]. Briefly, subjects chose between two visually presented lotteries on a touchscreen, each offering probabilistic gains or losses of tokens (−3 to +3) represented by pie charts, with probabilities ($p = 0.25, 0.50, 0.75, 1.00$) and outcomes encoded by distinct visual patterns (Fig. 1). The monkey has to select the most rewarding lottery among two displayed on the screen: $L_{right} = (x_{right}, p_{right})$ and $L_{left} = (x_{left}, p_{left})$. If the monkey chooses $L_{left}$ it will gain the quantity $x_{right}$ with a probability $p_{right}$ and similarly for the left side. After selecting a lottery, the outcome was resolved probabilistically, tokens were updated on a gauge, and liquid rewards were delivered proportionally to the final token count. The task includes different types of lotteries designed to test three conditions (Supplementary Fig. 10), both in the gain and loss domains: same-probability/different-outcome (quantity discrimination), same-outcome/different-probability (probability discrimination), and trade-offs between probability and outcome magnitude (risk-attitude assessment).

When facing uncertainty, rational behavior is defined as making choices that maximize the expected value[54] which can be expressed as follows:

$$EV = \sum_i P_i . x_i \qquad (1)$$

where $EV$ represents the expected value and $p_i$ the probability of obtaining outcome $x_i$. The probability weighting function $w(p)$ describes how people perceive the probability $P$. The function has an inverted S-shape, which illustrates that individuals tend to overestimate small probabilities, and

under-estimates big probabilities. However, the inverted S-shape is not a consistent feature[10,11]. The subjective utility function is defined as:

$$u(x) = \begin{aligned} & x^{1-\rho_+} \; if x > 0 \\ & = -\lambda(-x)^{1+\rho_-} \; if x < 0 \end{aligned} \qquad (2)$$

Where $u(x)$ represents the subjective utility of a quantity $x$, $\lambda$ is the loss aversion parameter. It controls the steepness of the curve in the loss domain, measuring the psychological impact of losses compared to gains. When $\lambda=1$, gains and losses are weighted the same. When $\lambda>1$, which is shown to be the most common behavior, people overweight losses compared to gains, making them loss-aversive. Conversely, when $\lambda<1$, people underweight losses compared to gains. $\rho+$ and $\rho-$ are the risk aversion parameters. They control the concavity and the convexity of the curve. If $\rho = 1$, indicating risk-neutral preferences. If $\rho < 1$, the curve is concave, indicating risk-aversive behavior. If $\rho > 1$, the curve is convex, indicating risk-seeking behavior. The existing literature has indicated that the shape of probability distortion varies across decision-making contexts, experimental settings, and individuals. Numerous formulations of the probability weighting function have been proposed, but no single unifying model has emerged[55]. Glöckner and Pachur found that simpler PT models with fewer free parameters tend to be more robust and less prone to overfitting[7]. Therefore, we selected the following formula to model probabilities weighting function:

**P1** (from ref. 56):

$$w(p) = exp(-(-ln\, p)^{\alpha}), \; \alpha > 0 \qquad (3)$$

We developed a Python framework to fit the PT models to the data. We model the monkeys' probability of choosing one option based on the difference in Subjective Expected Utility (SEV) between the two options. This is achieved using a sigmoid function:

$$P(L_{right}) = \frac{1}{1 + e^{\mu.(\Delta_{PT-x0})}} \qquad (4)$$

with $\Delta_{PT} = SEV_{right} - SEV_{left} = w(p_{right}).u(x_{right}) - w(p_{left}).u(x_{left})$

Where $p_{right}$, $p_{left}$ correspond to the probability of getting the non-zero outcome of lotteries right and left, respectively. $x_{right}$, $x_{left}$ correspond to the non-zero outcomes of lotteries right and left, respectively. $\mu$ regulates the steepness of the sigmoid curve, with well-trained monkeys showing a higher $\mu$. $x_0$ represents side bias: $x_0 > 0$ indicates a right-side bias, $x_0 < 0$, indicates a left-side bias. $\Delta_{PT}$ corresponds to the difference in SEV between the two lotteries. We ran a simulation in order to estimate the number of trials needed in order to estimate PT parameters using our gambling task. We found that after 1500 trials (any type of lotteries), the error does not decrease significantly (Supplementary Fig. 5). This measure has been averaged over 25 players (using random selection of value for the parameters).

### Data filtering

Data filtering was performed on $\rho$, $\alpha$, and $\lambda$ parameters from the full dataset of 1500 trial periods to exclude outlier PT parameters. To achieve this, the interquartile range (IQR) method was applied, removing parameters that fell below $Q1 - 1.5 \times IQR$ or above $Q3 + 1.5 \times IQR$. After this filtering, individuals who had less than 10 remaining 1500 trial periods were excluded. This procedure also helped eliminate incoherent parameter values, which could arise when a monkey consistently chose the option on only one side of the screen, possibly due to spatial biases or learned heuristics rather than genuine decision-making processes which may be linked to a form of status Quo bias[57]. As a result of this filtering method, six individuals were excluded, most of these individuals had a strong side bias ($74.96\% \pm 36.82$ same side choice) while the other individuals displayed a more acceptable side bias ($61.42\% \pm 12.16$). For the 18 remaining individuals, the filtering procedure discarded 16.94% of their trials. This results in

a dataset of 1,380,190 trials, collected over a period of 40 months (Supplementary Table 1).

## LMM modeling

Linear mixed-effect models (LMMs) were fitted to disentangle the impact of age, sex, and social rank (Elo-rating and COP) of individuals on PT parameters, which indicate risk attitudes, probability distortion, and loss aversion. Because they had several parameters for each 1500 trial periods, individuals were added as random factors to account for the nested nature of our data. All models were fitted on the dataset after data filtering. In addition, the mean number of trials per period was included as a covariate, as it was found to have a significant effect, likely reflecting an influence of experience on decision-making. Because age and mean trial number were collinear, we chose to categorize age into three categories: juveniles (individuals under 4 years old), subadults (4–8 years old for males, 4–7 years old for females), and adults (over 8 years old)[58]. A random intercept was included to account for the non-independence of data points, as each individual completed thousands of trials. Random slope was not included, as this would imply that all individuals respond similarly across the predictor, whereas in reality each individual was exposed to only a limited range of social ranks during data collection. All models were run under R version 4.3.1 within the Rstudio environment, using the "lmer" function of "lme4" package version 3.1-3 as follows:

$$PT\ parameter \sim age + sex + Elo\ score + Elo\ score^2 + trial\ number + (1 \mid individuals))$$

Each model hypothesis (e.g., normality of residuals, homoscedasticity, and linearity) were evaluated using the "check model" function from the "performance" package version 0.13.0. Due to the skewed distribution of the variables $\alpha$ and $\lambda$, a logarithmic transformation ($\log(\alpha)$ and $\log(\lambda)$) was applied prior to inclusion in the model to stabilize variance and meet model assumptions. A conservative threshold of $P < 0.01$ was used to reduce the risk of Type I errors. All statistical tests are two-sided.

## Reporting summary

Further information on research design is available in the Nature Portfolio Reporting Summary linked to this article.

## Data availability

The complete dataset containing all individual choices is publicly available at: https://doi.org/10.5281/zenodo.15090088.

## Code availability

All codes used to generate the main and Supplementary Figs. and tables can be accessed via: https://archive.softwareheritage.org/browse/origin/?origin_url=https://github.com/rougier/evoprospect-v2.

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

## Acknowledgements

The authors are grateful to the University of Strasbourg, the CNRS, and Silabe (silabe.com) for their support of this research and expert animal care. The development of the Machines for Automated Learning and Testing (MALT) was supported by the University of Strasbourg Institute for Advanced Study (USIAS) through a USIAS fellowship to Hélène Meunier, who also provided the image of the MALT apparatus in Fig. 1. We thank Adam Rimelé for technical and programming support of the MALT and Mathieu Lefebvre for his feedback on an earlier version of the manuscript. This work was funded by IDEX-ATT20 "PrimaRisk" and IDEX-RE23 "TOC" grants to S.B. N.C.E. was supported by the French Ministry of Higher Education and Research.

## Author contributions

S.B., T.B., N.P.R., and S.B.G. defined the question addressed and conceived the task. S.B. performed the data collection. N.C.E., N.P.R., S.B., and A.G. performed the analysis. All the authors contributed to the writing of the manuscript.

## Competing interests

The authors declare no competing interests.
