## [Transparent Peer Review file · Communications Biology]

Social hierarchy influences monkeys' risky decisions

Corresponding Author: Dr Sébastien Ballesta

Version 0:

Reviewer comments:

Reviewer #1

(Remarks to the Author)

This study investigates how social hierarchy influences risky decision-making in Tonkean macaques, using a large longitudinal dataset of over 1.3 million trials collected via an automated gambling task. The findings show that monkeys display decision biases consistent with Prospect Theory—risk aversion for gains, risk seeking for losses, and loss aversion. Importantly, social rank dynamically modulates these behaviors: middle-ranking individuals are less risk-averse for gains, potentially due to the social unpredictability they face. This relationship follows a U-shaped curve, where both low- and high-ranking monkeys show more predictable outcomes in social conflicts and more conservative decision-making patterns. The study demonstrates that risk attitudes are not fixed traits but adapt to changes in social context, particularly social status. Age and task experience also influence decision patterns, while sex has no significant effect. These results highlight the flexibility of cognitive biases in primates and provide valuable insights into how social structure shapes economic decisions across species.

This is a very strong and well-executed study, supported by an impressive volume of data and rigorous task design and statistical analyses. The methodology is sound, and the results are clearly presented and thoughtfully interpreted. My comments primarily focus on points of clarification and suggestions to enhance clarity for the reader.

1. If I understand correctly, the Elo-rating used to determine social hierarchy is based on displacement or conflict events occurring during access to the MALT device. Could these conflict interactions themselves influence the monkeys' performance on the cognitive task that immediately follows? It would be helpful if the authors could discuss whether the Elo-rating captures an individual's baseline social rank or reflects a dynamic, task-context-specific status. Is there a way to assess dominance rank independently of MALT-related interactions to ensure it reflects more naturalistic social hierarchy? Additionally, have the authors considered whether the outcome of a conflict (win vs. loss) has an acute impact on subsequent risk preferences? For instance, might middle-ranking individuals become more risk-seeking after a win and more risk-averse after a loss?

2. In Figure 2F, individuals are ordered by Elo-rating, and one might expect that the same U-shaped pattern of risk attitudes seen in Figure 3A (ρ for gains) would be visually reflected in the distribution of p values in Figure 2A. However, this pattern is not immediately apparent. Could the authors clarify this discrepancy? Is it due to averaging across trial periods or inter-individual variability that masks the overall trend?

3. Figure 3B–C shows that most monkeys are loss-averse ($\lambda > 1$) while also being risk-seeking in the loss domain ($\rho_- < 0$). This combination is intriguing, as it may suggest a compensatory cognitive mechanism—overweighting losses while simultaneously preferring risky options to avoid them. Do the authors have any thoughts on this?

Reviewer #2

(Remarks to the Author)

In this manuscript, Chaix-Eichel et al describe the effects of social status on risk-attitude in a group of macaques. They use an unusual and very rich data set: gambling task choices from a free-ranging group whose responses have been recorded over multiple years. This allows to observe changes in risk-attitude of individuals as their relative social rank varies in the group. Overall, I am very excited about this data set and about the manuscript. It shows clear and significant relationships of social rank and risky choice. I think there is a need for greater clarity in the presentation. I have listed my main concerns below. Once these issues are addressed, I think this will be an exciting paper.

Major concerns:

- (1) The main conclusion: 'middle-ranking Tonkean macaques exhibit reduced risk aversion in the gain domain, but this effect does not extend to the loss domain' is not true. In the loss domain, there is a significant modulation of the loss-aversion parameter, so that middle-ranking macaques have higher loss aversion (i.e., a value higher than 1). That should lead to higher risk-seeking in the loss domain. In other words, there is a general tendency to be more willing to accept risk in BOTH domains. The authors need to state this clearly. (Otherwise, I have completely misunderstood their behavioral findings.)
- (1) The Elo scores in Fig. 2 are somewhat ambiguous with respect to changes in hierarchy, largely, because they are shown in aggregate. It would be interesting to see them plotted across time for all monkeys. That would allow an assessment, how stable the hierarchy was and how often rank reversals were observed. [Later note: I see S3 shows this, which is good. A variant of this figure should be included in the main text. I would recommend showing the plots for the entire population and highlight two, or maybe three, individuals that show clear and consistent changes in rank.]
- (2) The effect of social rank on decisions under risk is interesting. The authors should make sure that this effect cannot be alternatively explained by sex, age, or (most likely) an interaction of sex x age. In particular, I would presume that social rank is less volatile in females than in males. Furthermore, in males social rank should first increase with age, reach a peak and then fall. Is that correct? I presume the authors use the LLM models to do this, but they do not explain the results with respect to the non-social factors in sufficient detail. At the very least they should show plots in the Suppl Material similar to the ones shown in Fig. 3 & 4 for the other variables.
- (3) Results, l.79ff: 'In summary, social hierarchy encompasses two distinct dynamics: a linear one that considers the individuals' rank and a U-shaped relationship reflecting more complex feature of social hierarchy.' This statement is either unclear or just wrong. I take it the authors refer to the linear ranking (Fig. 2F) and the U-shaped relationship of conflict outcome probability (Fig. 3F). But this later relationship is not 'a more complex feature' of the hierarchy, but simply a reflection of the fact that the monkeys on the low end of the hierarchy have MANY other monkeys that will win a conflict with them, and the opposite is true for the monkeys at the high end of the hierarchy. Only the monkeys in the middle have equal number of higher- and lower-ranking social partners, so that on average the conflict outcome is less certain. If the authors want to imply anything more complex than this, they should say so directly. Otherwise, they should simply state these regularities. BTW – how stable was the hierarchy at any given day (or maybe a longer time bin, 2-3 days)?
- (4) Results, l.86ff: Here the authors finally introduce the 1500 trial period aspect of the analysis. This description should appear at the very beginning of the result section. However, I am still confused about: (4.1) how are these trials chosen? (random sampling, consecutive trials); (4.2) how many trials have been recorded for each individual monkey, and in cases in which multiple sets of 1500 trials were recorded, where these additional data used and if yes in what form (e.g., blocks of trials that are close in time?); how many recording days equals 1500 trials, in other words how many trials did an individual monkey perform in a given day?
- (5) I do not completely understand the PT fit that the authors used. Specifically, they fitted 'risk attitude' separately for the gain and loss condition. This is fine, but I think this negates the simultaneous fit of a 'loss aversion' parameter. In the 'classical' PT model, a 'loss aversion' parameter is necessary, because the assumption is that the curvature of the utility function for gain and losses is just mirrored at the reference point. Thus, 'loss aversion' describes the increased sensitivity to losses, i.e. a different overall curvature of the loss utility function. However, in the modelling approach of the authors, the overall sensitivity of the utility function can be picked up either by the 'risk attitude in losses' and the 'loss aversion' parameter, or (most likely) by an unclear mixture of both parameters. I think the finding shown in Fig. 3, namely that in the loss domain, the 'risk attitude in losses' is mostly unaffected by rank (Fig. 3B). This is noticeably different from the 'risk attitude in gains' (Fig. 3A). Instead, the 'loss aversion' shows a strong relationship with rank. I wonder, if this is real or a byproduct of the fitting method. The authors should try two alternative fits: (5.1) They should 'clamp' the risk attitude parameters for gains and losses, so that they are different in sign, but equal in strength. One way of doing this would be to first fit the gain trials and then to use the 'risk attitude' parameter from that fit as an unfree parameter in the fit for loss trials. Now any difference in steepness of gain and loss utility functions can only be a result of the 'loss aversion' parameter. This would be most similar to 'classic PT'. (5.2) Alternatively, they could simply eliminate the 'loss aversion' parameter, so that the 'risk attitude in losses' fully reflects the overall utility function curvature. I wonder, if the parameter under these circumstances shows a relationship with rank.
- (6) If the findings of the authors are correct, they should be able to directly illustrate the effect using an individual monkey as an example by plotting the PT functions before and after a shift in rank. It is fine, if they cannot do this. In that case, I would recommend taking out references to rank changes of individual monkeys. They can make their point by referring to the static rank across the observed population and the observed relationship between rank and risky decision-making.
- (7) Going beyond the observation that middle-ranking monkeys vary from the low- and high-ranking monkeys in their choice behavior, it is of course of interest what might drive these differences. Here the authors speculate that (l. 149ff): 'Middle-ranking are more likely to encounter peers of that have a comparable Elo-rating which may trigger, competitive interaction are thus more frequent'. Here the authors do not need to speculate. They should be able to observe this hypothesized relationship directly in their behavioral data set. In this context, it would be extremely interesting, if winning or losing a competitive interaction has an effect on subsequent gamble choices. Are monkeys that just won a competitive interaction more or less risk-seeking?
- (8) I think the speculation in the end that early stages of the experiment are more similar to 'description-based gambles' and later ones to 'experience-based' is far-fetched. The authors have no independent measure of the learning system involved in the monkey's behavior or of the type of internal representation. It is therefore better to remove these speculations.

Minor concerns:

- (1) Results, l.48: Stating the size of the total number of trials is fine. How many individual monkeys were recorded? [I think 18 monkeys, but I got that from counting the data in Fig. 2. That is ridiculous.] What is their breakdown in terms of gender and age? How many trials are gathered for an individual monkey? These data should be presented immediately at the beginning of the results section, to inform the reader about the scope of the data set to be analyzed.
- (2) S1 & S2: The grey individual curves are close to undeliverable in my version of the figures. I would encourage you to

play around with different formats. At the very least, give the lines a higher contrast.

(3) S2: I think 'probability weighting function' is better nomenclature than 'probability distortion function', because it is more neutral.

(4) S2: While the individual functions for loss (yellow) are somewhat homogeneous and all have the general form of the average, the functions for gain (blue) seem to fall into two groups with symmetric, but inverse form. The resulting average is near meaningless, because it represents mostly the (presumably somewhat random) frequency of the two types in the population. It might be better to show the two groups separately. Incidentally, I would be curious, if the utility functions of these groups are similar or if they also show a difference.

(5) Figure 2: How can you have a distribution of parameter for each individual monkey? Is this across different recording period: days, month, years? This unclear from either the figure legend or the main text. It might be explained in the methods (I have not checked at the point, where I write this comment). However, even if, I am already annoyed. As a reader, I need to be informed about important aspects of the analysis as I encounter the figure, instead of flipping to the methods section and digging in it, until I find it.

(6) Figure 2: The 'grey points' are nearly indecipherable. (On my computer, I had to increase the size of the figure to 280%, before I could see some little points. This does not work.) Find a better way of showing this result, if it is important. What are the '1500 trial periods'? What is a 'trial period', why 1,500? Are they randomly drawn from a larger set, and if yes, why is not the larger set more informative?

(7) S3: The authors should make the grey lines higher in contrast and make all lines thicker. In addition, they should place marker (i.e., circles) each time a monkey rises or falls in rank consistently relative to another monkey. For me, it was not obvious that the monkeys often shifted rank. Instead, it seemed rather stable to me. The curves are all somewhat noisy. In that sense they are clearly 'dynamic', but it less clear that curves consistently rise or fall (except maybe 'Eri', 'Ces' and 'Bar'). Quantifying these events and indicating them in this figure would help.

(8) Figure 3: 'In all panels, each black point corresponds to a fitted social or decision-making parameters from 1500 trial periods per individual.' This description is at the very least ambiguous. Again, what are the '1500 trial periods per individual'. Furthermore, this sentence implies that each dot represents the analysis of 1,500 trials (or parts of trials, which would make no sense). However, you only have 18 monkeys, but you show 927 data points. So, each monkey provides multiple '1500 trial period' samples? If yes, how many? How are they chosen?

(9) Results, I.94: 'No significant correlation between Elo-rating and PT parameters was found in the loss domain, while all individuals displayed a comparable level of risk-seeking (LMM test, $p > 0.01$, Fig. 3B, Fig. 4A, Table 1).' This statement is incorrect. The authors should state that they talk about 'risk attitude in losses' parameters, not ALL PT parameters, which includes the 'loss aversion' parameter that shows a clear and significant inverse-U shape in its relationship with rank. The authors state this themselves a few lines later. This is just sloppy writing.

(10) Fig. 4: The authors should also show the 'loss aversion' parameter. I do not think this Fig. is important enough to be shown in the main text. The rank dynamic would be much more interesting.

(11) The plots from Fig. 3 should be shown separated by sex, and age-group (in Suppl. Material). This would nicely illustrate the absence of an influence on risky decision-making by these factors.

(12) It would be good to see the PT-fits (similar to S1) separately for three rank-groups (low, middle, high). That would show the combined effect on the utility functions and would clearly illustrate the behavioral effects.

Reviewer #3

(Remarks to the Author)

Animals' risk preferences affect their survival, and ultimately, their fitness. Although studies have explored risk preferences in non-human primates, whether risk preferences are modulated by factors such as sex and social rank has not been tested. In this original study, Chaix-Eichel and colleagues tackle this question by testing the risk preferences of Tonkean macaques living in a semi-free environment with an automated testing system. The authors discovered that risk preferences in these animals are not stable; rather, they depend on social hierarchy. These findings are important as they address a long-standing open question about the interplay of decision-making, and particularly risky decision-making, and its relationship to social dominance. We believe that the strength of the evidence behind these findings will be even more convincing, once the authors address our comments.

Main Comments

1. Methodologically, the study is observational; it is hard to draw a causal conclusion that the risk preference is not a stable trait but depends on social rank. For this, the authors used a regression including animals with both animals with stable social rank (the highest and the lowest ones) and animals with non-stable social rank. To what extent does this effect come from animals with stable social rank? Some analysis excluding these animals may help address this concern. Furthermore, it is unclear what Elo rating was used when correlating it to the decision parameters. The estimation of the decision parameter is based on 1500 trials, which we assume are completed in the span of several sessions in which the Elo rating might change. Also, Elo rating might not capture the relative rank of the animals. Would using relative rank instead of Elo ratings provide a different angle to the data? (We do not expect so, but it would be advantageous to know.)

2. The authors did not say how many kinds of trials they have for the lottery task. As indicated in Fig. 1, they can have $7 \times 2 + 7 \times 6/2 + 7 \times 6 = 77$ trial types, and they have to estimate 6 parameters. Even though the animals performed more than 1m decisions, simply repeating the same trial type does not guarantee the estimability of the parameters. The authors need to describe their trial types more explicitly to ensure that the parameters can be recovered from the data. Explicitly, write down all the lottery information in the methods, not only in the Figures. Furthermore, it will be advantageous to show that Tonkean macaques also show first-order stochastic dominance. That is, while holding reward probability or magnitude constant across choices, the animals chose the dominating option more often than chance.

3. Concerning the LMM model, the authors did not say why they chose this form of model. Such as, Why is only the elo-rating included as a second-order term? Why did the random effect only include a random intercept but not a random slope? The authors need to justify the model specification, perhaps by performing model selection.

Minor comments

1. Concerning the exclusion of animals, the authors excluded six animals from the raw data. They said the excluded animals have a severe side bias. We wonder if it is possible to incorporate this side bias into their model, and aid in the model selection process. This analysis can be complemented by showing that despite side biases (regardless of their magnitude), the animals chose the dominating option.

2. In the label of Fig. 1:

- a. We think it should read ...a 25% chance ... in the sentence "If the monkey chooses the left pie chart, it will have half a chance of getting 1 token. If it chooses the right pie chart, it will have a 75% chance to get 2 tokens."
- b. What is half a chance? Please revise.

3. In Figures 2 and 3, place the label, such as risk aversion, risk seeking, etc, near the axis.

4. It is unclear how using the MALT, which 'track(s) thousands of decisions over several years', enables daily updates of dominance hierarchies.

5. It's unclear how "Adopting the PT framework, we analyzed the decisions of 18 Tonkean macaques across 1,380,190 trials" allows the longitudinal study of risky decision making. Longitudinally would come from testing over time, but the sentence refers to the number of trials.

Raymundo Báez-Mendoza and Shen Zhang

Reviewer #4

(Remarks to the Author)

I co-reviewed this manuscript with one of the reviewers who provided the listed reports. This is part of the Communications Biology initiative to facilitate training in peer review and to provide appropriate recognition for Early Career Researchers who co-review manuscripts.

Version 1:

Reviewer comments:

Reviewer #1

(Remarks to the Author)

The authors have nicely addressed my comments, and the manuscript is much stronger now. I don't have further comments, and congratulations on the nice work!

Reviewer #3

(Remarks to the Author)

The authors have satisfactorily answered our comments. Nonetheless, we highly recommend including a statement in the methods justifying the use of ELO-ranking over relative rank. We also suggest that the authors include the results reported in Table R5 to the supplementary materials.

Reviewer #4

(Remarks to the Author)

I co-reviewed this manuscript with one of the reviewers who provided the listed reports. This is part of the Communications Biology initiative to facilitate training in peer review and to provide appropriate recognition for Early Career Researchers who co-review manuscripts.

We sincerely thank the reviewers for their thoughtful and constructive feedback on our manuscript. All reviewers expressed overall positive views, highlighting the relevancy, strength and rigorous execution of our study, as well as the substantial volume of data and the significance of the findings. We are also grateful for the detailed and thoughtful reviews that have helped us to improve the paper. We performed suggested additional analysis that helped us to better explain our findings, while leaving our core conclusions mostly unchanged. Figures and tables are included in our response document to clarify specific points. We chose not to include some of them in the final manuscript or its supplementary material to avoid overloading these sections and to stay focus on the scope of the actual study. In such cases, they are labeled as Figure or Table R + number.

We ensured that our revised manuscript follows all the guidelines from Communication biology.

Please find below, in blue, point by point answers to each reviewer's comment.

Reviewers' comments:

Reviewer #1 (Remarks to the Author):

This study investigates how social hierarchy influences risky decision-making in Tonkean macaques, using a large longitudinal dataset of over 1.3 million trials collected via an automated gambling task. The findings show that monkeys display decision biases consistent with Prospect Theory—risk aversion for gains, risk seeking for losses, and loss aversion. Importantly, social rank dynamically modulates these behaviors: middle-ranking individuals are less risk-averse for gains, potentially due to the social unpredictability they face. This relationship follows a U-shaped curve, where both low- and high-ranking monkeys show more predictable outcomes in social conflicts and more conservative decision-making patterns.

The study demonstrates that risk attitudes are not fixed traits but adapt to changes in social context, particularly social status. Age and task experience also influence decision patterns, while sex has no significant effect. These results highlight the flexibility of cognitive biases in primates and provide valuable insights into how social structure shapes economic decisions across species.

This is a very strong and well-executed study, supported by an impressive volume of data and rigorous task design and statistical analyses. The methodology is sound, and the results are clearly presented and thoughtfully interpreted. My comments primarily focus on points of clarification and suggestions to enhance clarity for the reader.

Thank you for your summary and positive views.

1. If I understand correctly, the Elo-rating used to determine social hierarchy is based on displacement or conflict events occurring during access to the MALT device.

Your understanding is correct: the Elo-rating used to determine social hierarchy is based on displacement or conflict events, specifically, one animal replacing another, at the MALT device or an automated testing station. Multiple studies mentioned in the article support the validity and reliability of this approach:

Gullstrand et al. (2021) show that the act of a baboon replacing another at a workstation were highly consistent with those from traditional, observer-based assessments of dominance hierarchies, with strong rank correlations, confirming that the automated method can reliably and efficiently capture the group's social hierarchy.

Ballesta et al. (2021). show that video analysis of MALT-based conflicts revealed that approximately 74-75% of such "replacements" reflected genuine social displacements indicating dominance. Hierarchies derived from these automated records showed a strong correlation (mean $r > 0.75$) with direct behavioral observations, supporting the robustness of using such automated devices for ranking individuals. Compared to traditional methods which require extensive observational labor, automated testing devices like MALT offer a powerful alternative or complement to direct behavioral observation for establishing social hierarchies in non-human primates.

Références:

Gullstrand, J., Claidière, N., & Fagot, J. (2021). Computerized assessment of dominance hierarchy in baboons (*Papio papio*). *Behavior Research Methods*, 53(5), 1923-1934

Ballesta, S., Sadoughi, B., Miss, F., Whitehouse, J., Aguenounon, G., & Meunier, H. (2021). Assessing the reliability of an automated method for measuring dominance hierarchy in non-human primates. *Primates*, 62(4), 595-607

Could these conflict interactions themselves influence the monkeys' performance on the cognitive task that immediately follows?

Gullstrand et al. (2021) directly addresses whether conflict interactions could affect the cognitive performance on the task that immediately follows. The study found that these supplanting events do, in fact, have a modest yet measurable impact: when a baboon supplanted another and took its place at the workstation, the supplanter typically showed a transient improvement in cognitive performance on the subsequent task. The boost in performance depended on the difference in dominance rank between the two baboons. So, the researchers concluded that the conflict or dominance interaction just before the test could temporarily influence immediate cognitive outcomes, with a short-term benefit observed for the baboon that gains access through supplanting. Marzouki et al. (2014) show that a baboon's *mood*, including the immediate aftereffects of conflict, can bias cognitive performance, specifically slowing down response speed after negative events.

In our dataset, the proportion of trials following a conflict to access MALT represent less than 1.5% of all trials. We therefore did not include such analysis in our manuscript as it is very unlikely that these trials would explain our findings. We however performed this analysis and a more detailed explanation of this analysis can be found below (Fig R1).

Référence:

Marzouki, Y., Gullstrand, J., Goujon, A., & Fagot, J. (2014). Baboons' response speed is biased by their moods. *PloS one*, 9(7), e102562

It would be helpful if the authors could discuss whether the Elo-rating captures an individual's baseline social rank or reflects a dynamic, task-context-specific status.

Is there a way to assess dominance rank independently of MALT-related interactions to ensure it reflects more naturalistic social hierarchy?

The Elo-rating derived from MALT-based conflict events predominantly reflects an individual's baseline social rank, not just a transient or task-specific status. Ballesta et al. (2021) demonstrated a strong correlation between dominance ranks inferred from MALT interactions and those obtained through traditional ethological observations. Video analyses confirmed that most displacement events at MALT (74.5%) were social displacements aligned with dominance behavior, while 16% were affiliative-like and 10% unclassifiable. To validate the ecological relevance of these rankings, Ballesta et al. cross-referenced MALT-based Elo-ranks with human-coded observations across a variety of natural social contexts. The close correspondence between the two methods confirms that MALT-derived hierarchies reflect broader, group-level social structures rather than being limited to device-specific interactions.

Additionally, have the authors considered whether the outcome of a conflict (win vs. loss) has an acute impact on subsequent risk preferences? For instance, might middle-ranking individuals become more risk-seeking after a win and more risk-averse after a loss?

In order to more precisely answer the reviewer comment, we performed the following additional analysis (Fig. R1). We first considered four conflict-related conditions: control (no conflict within the preceding 15 minutes), win (won a conflict within the preceding 15 minutes), loose (lost a conflict within the preceding 15 minutes), and both (experienced both a win and a loss within the preceding 15 minutes). Using a generalized linear mixed model (GLMM), we estimated the predicted probability of choosing the risky option across these conditions and task types (gains vs losses), and performed pairwise comparisons using Tukey-adjusted contrasts.

The results indicate that past conflict conditions can significantly influence attitude towards risk. Participants are substantially more likely to take risks in the loss domain compared to the gain domain. In contrast, recent conflict outcomes have only minimal effects.

In the gain domain, winning a conflict slightly increases risk-taking relative to control (odds ratio = 0.953, $p = 0.037$), losing a conflict shows a trend toward decreased risk (odds ratio = 1.051, $p = 0.076$), and the "both" condition is not significantly different from control (odds ratio = 0.995, $p = 0.997$). Pairwise comparisons among win, loose, and both show only one significant difference (win vs lose, odds ratio = 1.102, $p = 0.0015$).

In the loss condition, none of the contrasts reached significance, indicating that recent conflict outcomes do not meaningfully affect risk-taking when the task involves losses.

Overall, these analyses confirm that while some conflict-related effects in gains are statistically detectable, their magnitude is very small, and the "both" condition shows that experiencing both outcomes does not substantially alter risk behavior. Here, the type of task (gain vs loss)

remains the dominant factor shaping risk-taking. These results further validate one of our finding that risk attitude in the gain domains seems to be more flexible than in the loss domain.

Fig. R1. Probability of choosing risky options as a function of conflict status and task type.

The Y-axis represents the estimated probability based on conflict status, computed using the *emmeans* post-hoc test. Blue indicates gain trials, while orange indicates loss trials. A significant difference emerges only between the “win” and “lose” conflict statuses in the gain domain ($\chi^2 = 13.50, p = 0.0037$). No significant differences are observed in the loss domain ($\chi^2 = 1.67, p = 0.64$). The results indicate that task context is the primary driver of risk-taking: participants are substantially more likely to take risks in the losses task compared to the gains task (GLM, $p < 0.001$). In contrast, recent conflict outcomes have only minimal effects. In the gains condition, winning a conflict slightly increases risk-taking relative to control (odds ratio = 0.953, $p = 0.037$), losing a conflict shows a trend toward decreased risk (odds ratio = 1.051, $p = 0.076$), and the “both” condition is not significantly different from control (odds ratio = 0.995, $p = 0.997$). Pairwise comparisons among win, loose, and both show only one significant difference (win vs lose, odds ratio = 1.102, $p = 0.0015$).

2. In Figure 2F, individuals are ordered by Elo-rating, and one might expect that the same U-shaped pattern of risk attitudes seen in Figure 3A (ρ for gains) would be visually reflected in the distribution of ρ values in Figure 2A. However, this pattern is not immediately apparent. Could the authors clarify this discrepancy? Is it due to averaging across trial periods or inter-individual variability that masks the overall trend?

Before reviewing our response, please note that the figure you refer to as Figure 2 now corresponds to Figure 3, and the figure you refer to as Figure 3 now corresponds to Figure 4. These changes were necessary to accommodate an additional figure requested by the reviewers.

The fact that the described U-shaped distribution is more obvious in Figure 4 (former Figure 3) than in Figure 3 (former Figure 2) is mainly explained by the fact that latter shows the mean decisional attitude related to mean Elo-rating for each individual while the former reveals the decisional attitude for every 1500 trials periods. This supports the rationale of our analysis. Of note, in Figure 3 (former Figure 2), the U-shaped is more noticeable in the gains compared to loss domains (panels A and B, respectively) is also consistent with our analysis. Despite our subject inclusion criteria (see methods), individual ‘ber’ is an obvious outlier in panel A (Figure 3 and Figure S7), we believe that this can be explained by the fact that this female was either pregnant and after taking care of her baby during most of the data collection. We did not want to exclude this individual from our analysis or comment on this in the ms, as this issue is beyond the scope of this paper. In Fig. R.2, we also reproduce the same data as in Fig. 4A, showing only the median points used to fit the ρ parameter (without the violin plots), as this representation makes the U-shape pattern more apparent.

Fig. R2. Variability of ρ in gains for all individuals (same figure as Fig.3.A without the violin plots). The red points represent the mean points to fitted ρ parameters from 1500 trial periods per individual for gain trials.

3. Figure 3B–C shows that most monkeys are loss-averse ($\lambda > 1$) while also being risk-seeking in the loss domain ($\rho_- < 0$). This combination is intriguing, as it may suggest a compensatory

cognitive mechanism—overweighting losses while simultaneously preferring risky options to avoid them. Do the authors have any thoughts on this?

We agree that such a combination can appear to be intriguing. However, loss aversion and risk-seeking in the loss domain are distinct psychological phenomena and are not inherently incompatible, nor is their co-occurrence surprising. In fact, this combination is commonly observed in human behavior (Tversky, A., & Kahneman, D. (1992), Kahneman (2009)). Risk-seeking in the loss domain reflects a tendency to prefer options that offer a chance to minimize losses, even at the cost of greater variability. Intuitively, this aligns well with loss aversion, which refers to the tendency to weigh losses more heavily than equivalent gains. In other words, while individuals may be more sensitive to losses (loss aversion), they may also prefer to take risks to potentially avoid or reduce those losses (risk-seeking in loss). We added information on this point in the supplementary revised manuscript (see response to Reviewer #2, point 1(a))

Kahneman, D. (2009). The myth of risk attitudes. *Journal of Portfolio Management*, 36(1), 1

Tversky, A., & Kahneman, D. (1992). Advances in prospect theory: Cumulative representation of uncertainty. *Journal of Risk and uncertainty*, 5(4), 297-323.

Reviewer #2 (Remarks to the Author):

In this manuscript, Chaix-Eichel et al describe the effects of social status on risk-attitude in a group of macaques. They use an unusual and very rich data set: gambling task choices from a free-ranging group whose responses have been recorded over multiple years. This allows to observe changes in risk-attitude of individuals as their relative social rank varies in the group. Overall, I am very excited about this data set and about the manuscript. It shows clear and significant relationships of social rank and risky choice. I think there is a need for greater clarity in the presentation. I have listed my main concerns below. Once these issues are addressed, I think this will be an exciting paper.

Thank you for your thoughtful summary and positive feedback. We have made every effort to improve the clarity of the revised manuscript.

Major concerns:

(1a) The main conclusion: ‘middle-ranking Tonkean macaques exhibit reduced risk aversion in the gain domain, but this effect does not extend to the loss domain’ is not true. In the loss domain, there is a significant modulation of the loss-aversion parameter, so that middle-ranking macaques have higher loss aversion (i.e., a value higher than 1). That should lead to higher risk-seeking in the loss domain. In other words, there is a general tendency to be more willing to accept risk in BOTH domains. The authors need to state this clearly. (Otherwise, I have completely misunderstood their behavioral findings.)

In order to clarify this important aspect of the study, we ran additional analyses using alternative formulations of the Prospect Theory utility function: (i) one without the loss-aversion parameter (λ), and (ii) another that included λ but constrained the risk attitude parameter (ρ) to be the same for both gains and losses.

Model (i) without loss aversion (lambda)

$$u(x) = x^{1-\rho} \quad \text{if } x > 0$$
$$= (-x)^{1+\rho} \quad \text{if } x < 0$$

Model (ii) with loss aversion (lambda) but one parameter for risk attitude in loss and gains:

$$u(x) = x^{1-\rho} \quad \text{if } x > 0$$
$$= -\lambda(-x)^{1+\rho} \quad \text{if } x < 0$$

Overall, as depicted in figures R3-4 and tables R1-2, we still found a U-shape relationship of social hierarchy (Elo score) on risk attitude in the gain domain, but not in the loss domain.

We would like to emphasize that λ does not measure differences in risk attitude between gains and losses, but rather indicates the relative importance attributed to losses compared to gains in comparative choices. The effect observed on the loss-aversion parameter λ is most likely

driven by the fact that risk attitude shows a U-shaped relationship with Elo score in the gain domain, while it remains flat in the loss domain. Knowing that, we believe that our analysis supports the idea that we only observe the effect on hierarchy on risk attitude in the gain domains only.

Fig R3: Influence of social hierarchy on social and decision-making parameters using alternative PT models. In the utility function of this model, the loss-aversion parameter has been removed, making ρ the sole parameter used to describe risk attitude.

Predictors	rho_g			rho_l			alpha_g			alpha_l		
	Estimates	CI	p	Estimates	CI	p	Estimates	CI	p	Estimates	CI	p
(Intercept)	0.31	0.19 – 0.43	<0.001	-0.39	-0.49 – -0.28	<0.001	-0.16	-0.41 – -0.09	0.215	-0.51	-0.78 – -0.23	<0.001
age category [juvenile]	-0.13	-0.25 – -0.01	0.032	0.12	0.00 – 0.23	0.047	-0.59	-0.81 – -0.36	<0.001	-0.13	-0.44 – -0.18	0.402
age category [subadult]	-0.06	-0.10 – -0.01	0.010	-0.00	-0.04 – -0.04	0.989	0.04	-0.04 – -0.12	0.384	0.06	-0.05 – -0.17	0.286
sex [female]	0.11	-0.07 – 0.30	0.222	-0.02	-0.18 – -0.15	0.815	0.20	-0.19 – -0.60	0.310	-0.06	-0.49 – -0.37	0.782
elo	-0.05	-0.09 – -0.02	0.001	-0.06	-0.09 – -0.03	<0.001	0.02	-0.04 – -0.08	0.512	0.03	-0.05 – -0.11	0.505
elo^2	0.03	0.01 – 0.04	<0.001	-0.01	-0.02 – -0.01	0.337	-0.05	-0.08 – -0.02	0.001	0.02	-0.02 – -0.06	0.370
mean trial number	-0.11	-0.12 – -0.10	<0.001	0.06	0.05 – 0.08	<0.001	0.02	-0.01 – -0.04	0.234	0.01	-0.02 – -0.05	0.442
Random Effects												
σ^2	0.02			0.02			0.08			0.15		
τ_{00}	0.04	monkey		0.03	monkey		0.17	monkey		0.19	monkey	
ICC	0.61			0.57			0.67			0.55		
N	18	monkey		18	monkey		18	monkey		18	monkey	
Observations	962			962			962			962		
Marginal R ² / Conditional R ²	0.338 / 0.741			0.091 / 0.611			0.103 / 0.708			0.015 / 0.553		

Table R1. LMM models results for PT parameters using the model without the loss-aversion parameter. All models were computed on all 1500 trials periods for all individuals after filtering data. Values in bold represent significant effects ($p < 0.05$).

Model (ii) with loss aversion but same rho for gain/loss:

Fig R4: Influence of social hierarchy on social and decision-making parameters using alternative PT models. In the utility function of this model, the risk attitude parameter has been fixed for gains and losses, with free loss-aversion parameter.

Predictors	rho			alpha			lambda_		
	Estimates	CI	p	Estimates	CI	p	Estimates	CI	p
(Intercept)	-0.55	-0.75 – -0.36	<0.001	0.74	0.59 – 0.89	<0.001	1.85	1.26 – 2.44	<0.001
age category [juvenile]	0.25	0.06 – 0.45	0.010	-0.20	-0.35 – -0.04	0.014	-0.87	-1.49 – -0.24	0.007
age category [subadult]	0.09	0.02 – 0.16	0.013	0.07	0.02 – 0.13	0.012	-0.37	-0.60 – -0.14	0.002
sex [female]	0.16	-0.12 – 0.45	0.263	-0.05	-0.27 – 0.17	0.661	-0.45	-1.32 – 0.42	0.315
elo	-0.04	-0.09 – 0.01	0.087	-0.01	-0.05 – 0.03	0.642	0.03	-0.13 – 0.19	0.710
elo^2	0.07	0.04 – 0.09	<0.001	-0.03	-0.05 – -0.01	0.002	-0.23	-0.31 – -0.16	<0.001
mean trial number	-0.04	-0.06 – -0.02	0.001	0.03	0.01 – 0.05	<0.001	0.09	0.02 – 0.16	0.019
Random Effects									
σ^2	0.06			0.04			0.68		
τ_{00}	0.09	monkey		0.05	monkey		0.85	monkey	
ICC	0.60			0.56			0.55		
N	19	monkey		19	monkey		19	monkey	
Observations	1039			1039			1039		
Marginal R ² / Conditional R ²	0.150 / 0.659			0.079 / 0.595			0.108 / 0.603		

Table R2. LMM models results for PT parameters using the model with the loss-aversion parameter but fixed risk attitude parameter for gains and losses. All models were computed on all 1500 trials periods for all individuals after filtering data. Values in bold represent significant effects ($p < 0.05$).

(1b) The Elo scores in Fig. 2 are somewhat ambiguous with respect to changes in hierarchy, largely, because they are shown in aggregate. It would be interesting to see them plotted across time for all monkeys. That would allow an assessment, how stable the hierarchy was and how often rank reversals were observed. [Later note: I see S3 shows this, which is good. A variant of this figure should be included in the main text. I would recommend showing the plots for the entire population and highlighting two, or maybe three, individuals that show clear and consistent changes in rank.]

We thank the reviewer for this helpful comment and have included the corresponding figure below in the revised manuscript:

Figure 2. Evolution of Elo-rating for all individuals. Each panel's title shows the individual's name, age, sex, mean Elo-rating and total number of trials. Blue lines indicate the daily Elo-rating of the individual, while grey lines show those of others. Red dashed lines mark the first and last economic trials (training phases excluded). The larger panel (bottom right) highlights three individuals with distinct trajectories: Pac and Las display fluctuations with increases and

decreases in Elo-ratings, whereas Ces shows a steady upward trend reflecting his rise in hierarchy.

(2) The effect of social rank on decisions under risk is interesting. The authors should make sure that this effect cannot be alternatively explained by sex, age, or (most likely) an interaction of sex x age. In particular, I would presume that social rank is less volatile in females than in males. Furthermore, in males social rank should first increase with age, reach a peak and then fall. Is that correct? I presume the authors use the LLM models to do this, but they do not explain the results with respect to the non-social factors in sufficient detail. At the very least they should show plots in the Suppl Material similar to the ones shown in Fig. 3 & 4 for the other variables.

We thank the reviewer for this insightful comment regarding potential confounding effects of sex, age, and their interaction on the relationship between social rank and decision-making under risk. In the revised version of the manuscript, we have added supplementary figures showing the relationships between Age, Sex/PT, and Elo scores (see new Supplementary Fig. S9 and S10).

Regarding interactions between sex and age, we acknowledge their potential influence as well as ethological relevancy, however, due to the moderate sample size and the non-homogeneous sampling of our study (in particular, older males are frequently removed from the social group when social instability arises due to competition between adult males; see Ballesta et al. 2021 for examples), we are unable to reliably model these interactions. For this reason, we chose to present the effects of age and sex separately rather than including an interaction term.

Fig S9: Influence of age category on social and decision-making parameters. In all panels, each point corresponds to the mean of all fitted social or decision-making parameters from 1500 trial periods per individual. The darker points represent males, whereas the lighter points represent females. Blue points correspond to PT parameters in the gain domain, while orange points correspond to those in the loss domain.

Fig S10: Influence of sex on social and decision-making parameters. In all panels, each point represents the mean of all fitted social or decision-making parameters from 1,500 trial periods per individual. Darker points represent adults, while lighter points represent juveniles, with subadults shown in intermediate blue. Blue points correspond to PT parameters in the gain domain, whereas orange points correspond to parameters in the loss domain.

(3) Results, l.79ff: 'In summary, social hierarchy encompasses two distinct dynamics: a linear one that considers the individuals' rank and a U-shaped relationship reflecting more complex features of social hierarchy.' This statement is either unclear or just wrong. I take it the authors refer to the linear ranking (Fig. 2F) and the U-shaped relationship of conflict outcome probability (Fig. 3F). But this later relationship is not 'a more complex feature' of the hierarchy, but simply a reflection of the fact that the monkeys on the low end of the hierarchy have MANY other monkeys that will win a conflict with them, and the opposite is true for the monkeys at the high end of the hierarchy. Only the monkeys in the middle have an equal number of higher- and lower-ranking social partners, so that on average the conflict outcome is less certain. If the authors want to imply anything more complex than this, they should say so directly.

We apologize for the misunderstanding. We referred to the work 'complex' to describe the mathematical relationship (modelling U-shape relationships is arguably more 'complex' than a linear ones) and not the behavioral one. We agreed that our phrasing was unclear, so we removed the word 'complex' and changed the phrasing of our sentence.

"In summary, social hierarchy encompasses two related but distinct patterns: a linear ranking of individuals in the social hierarchy, and a U-shaped relationship reflecting the conflict

outcome probabilities, where low and high-ranking monkeys have more predictable results, and mid-ranking individuals face more balanced, uncertain competitions.”

Otherwise, they should simply state these regularities. BTW – how stable was the hierarchy at any given day (or maybe a longer time bin, 2-3 days)?

To address this, we calculated the mean standard deviation of Elo-scores across all individuals for different time windows (3 days to 1 year, Table R3).

Our results show that the hierarchy is relatively stable over short time windows: the mean standard deviation of Elo-scores is low at 3 days (3.45) and 7 days (5.68), with average Elo-scores ranging between ~915-925. As the time window increases, variability naturally grows (mean SD = 12.13 at 30 days; 32.06 at 6 months; 46.50 at 1 year), reflecting long-term changes in the social structure.

Thus, over the 2–3-day or 1-week time bins, the hierarchy remains highly stable, with only minimal fluctuations in Elo-scores. The higher variability observed at longer time scales likely reflects turnover events and social reorganization rather than short-term instability.

Elo-score global variability

Window	Mean standard deviation	Average minimum	Average maximum
3j	3.45	918.55	925.00
7j	5.68	914.20	929.04
30j	12.13	900.03	941.61
6m	32.06	853.47	977.80
1y	46.50	820.24	1,003.42

Table R3: Mean standard deviation of Elo-score for all individuals and for different time lines. The mean standard deviation was calculated from the Elo-score evolution of all individuals, and the minimum and maximum averages are provided here for reference.

(4) Results, l.86ff: Here the authors finally introduce the 1500 trial period aspect of the analysis. This description should appear at the very beginning of the result section. However, I am still confused about:

(4.1) how are these trials chosen? (random sampling, consecutive trials);

The 1500 trials were consecutive. A clearer explanation of the 1500-trial period is now provided at the beginning of the Results section, in the subsection “Social hierarchy influences risk attitude for gains.” We wrote:

“To dynamically assess the effect of changes in social hierarchy on economic decision-making, we segmented each monkey’s trial sequence into consecutive periods of 1,500 decisions. This window size corresponds to the estimated number of trials required to reliably compute PT parameters (**Fig. S6**). This segmentation enabled us to track how PT parameters evolved across successive periods.”

(4.2) how many trials have been recorded for each individual monkey, and in cases in which multiple sets of 1500 trials were recorded, where these additional data used and if yes in what form (e.g., blocks of trials that are close in time?); how many recording days equals 1500 trials, in other words how many trials did an individual monkey perform in a given day?

A new table has been added summarizing the trial numbers and participants’ ages for all categories of individuals (see response to minor comment 1). The PT parameters were assessed for all individuals over approximately 1500 trial periods, allowing us to observe the evolution of economic behavior over time. Below, we provide the distribution of days per 1500 trial periods and the distribution of trials per day.

Fig. S1. (A) Distribution of the number of days within a 1500-trial period. For clarity, the x-axis is limited to 50 days in the blue panel, which encompasses approximately 95% of all periods included in the analyses. Periods shorter than 10 days account for 66.34% of all periods analyzed. **(B) Distribution of the number of trials per day.** Red points indicate the mean number of trials per day for each individual. Fewer than 50 trials per day represent 31.80% of the dataset, fewer than 100 trials account for 46.88%, fewer than 250 trials for 71.75%, fewer than 500 trials for 92.69%, and fewer than 1000 trials for 98.92% of the dataset.

(5) I do not completely understand the PT fit that the authors used. Specifically, they fitted 'risk attitude' separately for the gain and loss condition. This is fine, but I think this negates the simultaneous fit of a 'loss aversion' parameter. In the 'classical' PT model, a 'loss aversion' parameter is necessary, because the assumption is that the curvature of the utility function for gain and losses is just mirrored at the reference point. Thus, 'loss aversion' describes the increased sensitivity to losses, i.e. a different overall curvature of the loss utility function. However, in the modelling approach of the authors, the overall sensitivity of the utility function can be picked up either by the 'risk attitude in losses' and the 'loss aversion' parameter, or (most likely) by an unclear mixture of both parameters. I think the finding shown in Fig. 3, namely that in the loss domain, the 'risk attitude in losses' is mostly unaffected by rank (Fig. 3B). This is noticeably different from the 'risk attitude in gains' (Fig. 3A). Instead, the 'loss aversion' shows a strong relationship with rank. I wonder, if this is real or a byproduct of the fitting method. **The authors should try two alternative fits: (5.1) They should 'clamp' the risk attitude parameters for gains and losses, so that they are different in sign, but equal in strength.** One way of doing this would be to first fit the gain trials and then to use the 'risk attitude' parameter from that fit as an unfree parameter in the fit for loss trials. Now any difference in steepness of gain and loss utility functions can only be a result of the 'loss aversion' parameter. This would be most similar to 'classic PT'. (5.2) **Alternatively, they could simply eliminate the 'loss aversion' parameter, so that the 'risk attitude in losses' fully reflects the overall utility function curvature. I wonder, if the parameter under these circumstances shows a relationship with rank.**

Before reviewing our response, please note that the figure you refer to as Figure 3 now corresponds to Figure 4. These changes were necessary to accommodate an additional figure requested by the reviewers.

We conducted these additional analyses and obtained similar results. Please refer to our response to comment Reviewer#2 (1a) and the corresponding figures for details.

In summary, we performed supplementary analyses using alternative formulations of the Prospect Theory utility function: (i) one excluding the loss-aversion parameter (λ), and (ii) another including λ but constraining the risk attitude parameter (ρ) to be identical for gains and losses. We still found a U-shape relationship of social hierarchy (Elo score) on risk attitude in the gain domain, but not in the loss domain.

(6) If the findings of the authors are correct, they should be able to directly illustrate the effect using an individual monkey as an example by plotting the PT functions before and after a shift in rank. It is fine, if they cannot do this. In that case, I would recommend taking out references to rank changes of individual monkeys. They can make their point by referring to the static rank across the observed population and the observed relationship between rank and risky decision-making.

Unfortunately, as individuals do not transition from the lowest to the highest rank over their lifetime, we do not expect to find a statistically significant U-shaped relationship between Elo-rating and risk attitude at individual level. That being said, Figure S7 illustrates these individual relationships can indeed be noticed that (i) each monkey spans only a limited portion of the Elo-score range throughout the study period and (ii) individual linear relationship somehow follows trends that are consistent with our finding as individual slopes seem negative in low Elo-scores, flat in the middle, and slightly positive in high Elo-scores.

(7) Going beyond the observation that middle-ranking monkeys vary from the low- and high-ranking monkeys in their choice behavior, it is of course of interest what might drive these differences. Here the authors speculate that (l. 149ff): 'Middle-ranking are more likely to encounter peers that have a comparable Elo-rating which may trigger, competitive interactions are thus more frequent'. Here the authors do not need to speculate. They should be able to observe this hypothesized relationship directly in their behavioral data set.

The hypothesis is indeed supported by our data and is illustrated in Figure 4F, which shows the relationship between conflict outcome predictability (COP) and Elo-rating. As described in the Results section, low- and high-ranking individuals display high COP scores, indicating more predictable outcomes in future conflicts. In contrast, middle-ranking individuals exhibit lower COP, reflecting greater uncertainty in conflict resolution, presumably due to more frequent encounters with similarly ranked peers. Thus, middle-ranking people are more likely to encounter peers that have a comparable Elo-rating, triggering more frequent competitive interactions, which induce behavioral and physiological changes that influence cognitive processes involved in risk-taking.

In this context, it would be extremely interesting, if winning or losing a competitive interaction has an effect on subsequent gamble choices. Are monkeys that just won a competitive interaction more or less risk-seeking?

We thank the reviewer for this relevant comment. Please refer to answer of Reviewer #1 that raised a similar point. In a nutshell, past conflict has been found to have a small effect on risk-attitude. Since, trials close to conflict to access to the MALT only represent 1.5% of all trials, we consider this as a second order effect that may be further explored in a separate study.

(8) I think the speculation in the end that early stages of the experiment are more similar to 'description-based gambles' and later ones to 'experience-based' is far-fetched. The authors have no independent measure of the learning system involved in the monkey's behavior or of the type of internal representation. It is therefore better to remove these speculations.

We agree with the reviewer's comment and removed this speculation from the discussion of the revised ms. Specifically, we removed this paragraph:

"Experience with the decision-making task significantly influenced risk attitudes. This result aligns with studies highlighting a description–experience gap in economic decision-making. Research suggests that description-based methods (where probabilities are clearly known) tend to lead to risk aversion or less risk-seeking in the gain domain, whereas experience-based methods (where probabilities must be learned trial by trial) are more often linked to risk-seeking behavior. In this sense, our longitudinal study may be interpreted as descriptive in the initial trials, gradually transitioning into an experience-based approach as monkeys accumulate more trials. Interestingly, this effect seems to be present in the gain but not in the loss domain. This further suggests that primates' cognitive biases are not rigid but can adapt based on task familiarity, reflecting a form of cognitive flexibility previously underappreciated in non-human species. Notably, probability distortion remained unaffected by experience, indicating that some cognitive biases may be more resistant to experience than others. "

and replaced it by:

"Finally, the number of trials completed by an individual significantly influences their risk behavior when making decisions involving gains. The more experience the individual accumulates with the task, the less risk-averse they become. Conversely, in the loss domain, increased task exposure leads to a reduction in risk-taking behavior, although this effect is less pronounced than in the gain domain. This pattern suggests a greater flexibility in decision-making processes related to gains, as opposed to the more stable or rigid patterns observed in losses, highlighting potential differences in the underlying cognitive and affective mechanisms involved in processing gains versus losses."

Minor concerns:

(1) Results, 1.48: Stating the size of the total number of trials is fine. How many individual monkeys were recorded? [I think 18 monkeys, but I got that from counting the data in Fig. 2. That is ridiculous.] What is their breakdown in terms of gender and age? How many trials are gathered for an individual monkey? These data should be presented immediately at the beginning of the results section, to inform the reader about the scope of the data set to be analyzed.

The number of monkeys is written in the Materials and Methods and in the Results. We added a supplementary table including all of this information.

Trials Summary by Sex and Age Category					
	N	Total Trials	Trials per Individual	Trials per Day	Age
Females					
Juvenile	1	9990	9990.0 ± NA	136.85 ± NA	3.6 ± NA
Subadult	3	122190	40730.0 ± 23866.3	151.59 ± 30.81	5.9 ± 1.3
Adult	5	277001	55400.2 ± 48784.3	146.10 ± 62.34	13.1 ± 6.1
Males					
Juvenile	2	39555	19777.5 ± 4635.1	185.91 ± 73.21	2.7 ± 0.1
Subadult	6	427862	71310.3 ± 68188.1	276.77 ± 89.87	6.3 ± 0.8
Adult	8	504230	63028.8 ± 51674.3	175.56 ± 76.53	8.3 ± 1.3

Table S2 : Summary of analyzed data by sex and age category of individuals

(2) S1 & S2: The grey individual curves are close to undeliverable in my version of the figures. I would encourage you to play around with different formats. At the very least, give the lines a higher contrast.

Thank you for your suggestion. We have implemented this change in the revised manuscript, and is shown below.

Fig. S3. Utility functions illustrating risk attitudes across domains for three Elo-score categories. Monkeys are grouped by Elo score: high (left), middle (center), and low (right). Elo ratings were categorized by dividing the overall range of the group's Elo scores into three equal tiers: low (584.12 to 835.81), middle (835.81 to 1087.5), and high (1087.5 to 1339.19). The blue curve represents the mean utility in the gain domain, while the orange curve represents the loss domain. Black curves depict individual utility functions. Overall, the utility function is concave in the gain domain, reflecting general risk aversion, and convex in the loss domain, indicating risk seeking. The steeper slope in the loss domain compared to the gain

domain highlights an overall tendency toward loss aversion. Mean parameter values are indicated in the bottom-left corner.

Fig. S4. Probability weighting function across domains for three Elo-score categories. Monkeys are grouped by Elo score: high (left), middle (center), and low (right). Elo ratings

were categorized by dividing the overall range of the group's Elo scores into three equal tiers: low (584.12 to 835.81), middle (835.81 to 1087.5), and high (1087.5 to 1339.19). Blue and orange curves represent overall gain and loss probability distortion respectively while grey curves represent the mean probability distortion for each individual. Top: The blue curves represent probability weighting function for gain. The distortion parameter α_+ is superior to 1 indicating a global and overall, under estimation of low probabilities and over estimation of high probabilities for gains. Bottom: The orange curves represent probability weighting function for gain. The distortion parameter α_- is inferior to 1 indicating a global and over estimation of low probabilities and under estimation of high probabilities for losses.

(3) S2: I think 'probability weighting function' is better nomenclature than 'probability distortion function', because it is more neutral.

We thank the reviewer for the correction; this has been changed in the manuscript when relevant.

(4) S2: While the individual functions for loss (yellow) are somewhat homogeneous and all have the general form of the average, the functions for gain (blue) seem to fall into two groups with symmetric, but inverse form. The resulting average is near meaningless, because it represents mostly the (presumably somewhat random) frequency of the two types in the population. It might be better to show the two groups separately. Incidentally, I would be curious, if the utility functions of these groups are similar or if they also show a difference.

Find bellow the suggested analysis depicted in Figure R5. Based on our analysis, we believe that utility functions, particularly in the gain domain, can vary within an individual, and that part of this variability is related to social rank and experience. In addition, because of the diversity in terms of age, sex, experience and social dynamics of the subjects, we feel it may not be ideal to group individuals solely based on their average utility function. We hope this clarification helps to contextualize our approach.

Fig R5: Utility function for two groups with different probability weighting function parameters in gains. We separated the individuals into two groups based on their probability weighting parameter (α) in the gain domain: one with $\alpha > 1$ and another with $\alpha < 1$. Overall, there does not appear to be a substantial difference in utility within the gain domain between these two groups (all are risk-averse). However, individuals with $\alpha < 1$, who underweight small probabilities and overweight larger ones, tend to be more risk-averse, which aligns with theoretical expectations.

(5) Figure 2: How can you have a distribution of parameters for each individual monkey? Is this across different recording period: days, month, years? This unclear from either the figure legend or the main text. It might be explained in the methods (I have not checked at the point, where I write this comment). However, even if, I am already annoyed. As a reader, I need to be informed about important aspects of the analysis as I encounter the figure, instead of flipping to the methods section and digging in it, until I find it.

(6) Figure 2: The 'grey points' are nearly indecipherable. (On my computer, I had to increase the size of the figure to 280%, before I could see some little points. This does not work.) Find a better way of showing this result, if it is important. What are the '1500 trial periods'? What is a 'trial period', why 1,500? Are they randomly drawn from a larger set, and if yes, why is not the larger set more informative?

Answer to (5) and (6) :

A clearer explanation of the 1500-trial period is now provided at the beginning of the Results section, in the subsection "Social hierarchy influences risk attitude for gains." We wrote: "To dynamically assess the effect of changes in social hierarchy on economic decision-making, we segmented each monkey's trial sequence into consecutive periods of 1,500 decisions. This window size corresponds to the estimated number of trials required to reliably compute PT parameters (Fig. S6). This segmentation enabled us to track how PT parameters evolved across successive periods."

A new table (Table S1) has been added summarizing the number of trials and participants' ages across all categories of individuals (see response to Minor Comment 1). The PT parameters were estimated for each individual over periods of approximately 1500 trials, allowing us to examine the evolution of economic behavior over time. In addition, we reported the distribution of days within these 1500-trial periods as well as the distribution of trials per day in a new Figure S1.

(7) S3: The authors should make the grey lines higher in contrast and make all lines thicker. In addition, they should place markers (i.e., circles) each time a monkey rises or falls in rank consistently relative to another monkey. For me, it was not obvious that the monkeys often shifted rank. Instead, it seemed rather stable to me. The curves are all somewhat noisy. In that sense they are clearly 'dynamic', but it less clear that curves consistently rise or fall (except maybe 'Eri', 'Ces' and 'Bar'). Quantifying these events and indicating them in this figure would help.

In this study, we did not consider the dynamic of Elo-rating at this level. Our analysis considers the mean Elo-rating for a period corresponding to 1500 trials. Considering the rise and falls of Elo-rating for each individual would represent a non-trivial detection of breaking point in Elo-rating and would imply to tackle fascinating questions related to how monkeys 'feels' when gaining or losing social status. In fact, we should discuss whether individuals are 'aware' that their social status is changing. In addition, potential results could be explained by confounding factors such as the number of defeats or victories experienced by each individual during such rises or falls. While we agree that this line of research is promising, we believe that the present study already provides a solid foundation and set of insights, even without addressing these transient and more complex effects of social dynamics on individual cognitive and emotional states.

(8) Figure 3: 'In all panels, each black point corresponds to a fitted social or decision-making parameters from 1500 trial periods per individual.' This description is at the very least ambiguous. Again, what are the '1500 trial periods per individual'. Furthermore, this sentence implies that each dot represents the analysis of 1,500 trials (or parts of trials, which would make no sense). However, you only have 18 monkeys, but you show 927 data points. So, each monkey provides multiple '1500 trial period' samples? If yes, how many? How are they chosen?

We apologize for any confusion. As mentioned above, we have provided a clearer explanation of the 1,500-trial periods at the beginning of the Results section. In addition, we have included the distribution of days per 1,500-trial period as well as the distribution of trials per day in Figure S1).

(9) Results, I.94: 'No significant correlation between Elo-rating and PT parameters was found in the loss domain, while all individuals displayed a comparable level of risk-seeking (LMM test, $p > 0.01$, Fig. 3B, Fig. 4A, Table 1).' This statement is incorrect. The authors should state that they talk about 'risk attitude in losses' parameters, not ALL PT parameters, which includes the 'loss aversion' parameter that shows a clear and significant inverse-U shape in its

relationship with rank. The authors state this themselves a few lines later. This is just sloppy writing.

The risk-attitude PT parameters have been specified in the text, and we acknowledge that this may have caused some confusion. It is important to clarify, however, that loss aversion refers not to the absolute gain or loss domain, but to the relative weighting of losses versus gains in comparative choices, in accordance with the original Prospect Theory framework (see our response to Reviewer #1, point 1a, regarding the loss-aversion parameter).

(10) Fig. 4: The authors should also show the ‘loss aversion’ parameter. I do not think this Fig. is important enough to be shown in the main text. The rank dynamic would be much more interesting.

We thank the reviewer for this suggestion. The loss-aversion parameter has been added to the figure, which has been relocated to the Supplementary Material as Figure S6, while the rank dynamics figure has been moved to the main text as Figure 2.

Fig S6. LMM predictions for ρ and α as a function of Elo score and mean trial number. (A). Predicted values for ρ for gains (blue) and losses (orange) as a function of Elo score. (B). Predicted values for α for gains and losses as a function of Elo score. (C). Predicted values for λ as a function of Elo score. (D). Predicted values for ρ for gains and losses as a function of mean trial number, a proxy of subject experience in the task. (E). Predicted values for α for gains and losses as a function of mean trial number. (F). Predicted values for λ as a function of mean trial number. In all panels, predicted values parameters ρ and α are plotted separately for the gains condition (blue) and the losses condition (orange). The shaded areas around the regression lines represent the 95% confidence intervals, providing an estimate of the uncertainty around the predicted values. Estimates and CI are provided by LMM from table 1.

Solid lines indicate statistically significant effects ($p < 0.01$), while dashed lines indicate non-significant effects ($p > 0.01$).

(11) The plots from Fig. 3 should be shown separated by sex, and age-group (in Suppl. Material). This would nicely illustrate the absence of an influence on risky decision-making by these factors.

Before reviewing our response, please note that the figure you refer to as Figure 3 now corresponds to Figure 4.

We thank the reviewer for this suggestion. We have implemented it in the new Figures S9 and S10 (see also our response to Comment 2).

(12) It would be good to see the PT-fits (similar to S1) separately for three rank-groups (low, middle, high). That would show the combined effect on the utility functions and would clearly illustrate the behavioral effects.

Following your suggestion, the PT fits are now shown separately for the three Elo-score categories in Figures S3 and S4 (see also our response to Minor Concerns 2–3).

Reviewer #3 (Remarks to the Author):

Animals' risk preferences affect their survival, and ultimately, their fitness. Although studies have explored risk preferences in non-human primates, whether risk preferences are modulated by factors such as sex and social rank has not been tested. In this original study, Chaix-Eichel and colleagues tackle this question by testing the risk preferences of Tonkean macaques living in a semi-free environment with an automated testing system. The authors discovered that risk preferences in these animals are not stable; rather, they depend on social hierarchy. These findings are important as they address a long-standing open question about the interplay of decision-making, and particularly risky decision-making, and its relationship to social dominance. We believe that the strength of the evidence behind these findings will be even more convincing, once the authors address our comments.

Thank you for your summary and overall positive views.

Main Comments

1. Methodologically, the study is observational; it is hard to draw a causal conclusion that the risk preference is not a stable trait but depends on social rank. For this, the authors used a regression including animals with both animals with stable social rank (the highest and the lowest ones) and animals with non-stable social rank. To what extent does this effect come from animals with stable social rank? Some analysis excluding these animals may help address this concern.

We repeated the analysis including only monkeys with a clearly non-stable social rank, specifically “eri,” “ces,” “fic,” and “bar.” The results are depicted in Table R4, and show that Elo (social hierarchy) has a significant effect on risk attitude, with a pattern consistent with our previous findings. Therefore, our finding cannot be solely explained by differences in stability of social rank between individuals.

Predictors	ρ^+			ρ^-			α^+			α^-			λ		
	Estimates	CI	p	Estimates	CI	p	Estimates	CI	p	Estimates	CI	p	Estimates	CI	p
(Intercept)	-0.05	-0.21 - 0.12	0.574	-0.54	-0.61 - -0.46	<0.001	0.30	-0.05 - 0.65	0.088	0.02	-0.20 - 0.24	0.856	0.96	0.60 - 1.32	<0.001
age category [juvenile]	0.04	-0.20 - 0.28	0.748	0.24	0.11 - 0.37	<0.001	-1.31	-1.62 - -1.01	<0.001	-0.37	-0.61 - -0.13	0.003	-0.70	-1.19 - -0.21	0.005
age category [subadult]	-0.09	-0.22 - 0.03	0.143	-0.01	-0.08 - 0.05	0.703	-0.14	-0.30 - 0.01	0.074	-0.21	-0.34 - -0.09	0.001	0.09	-0.15 - 0.34	0.458
sex [female]	-0.39	-0.61 - -0.18	<0.001	0.09	0.01 - 0.16	0.028	0.98	0.36 - 1.60	0.002	0.12	-0.23 - 0.48	0.498	0.76	0.23 - 1.29	0.005
elo	-0.14	-0.21 - -0.07	<0.001	-0.04	-0.08 - -0.01	0.007	0.06	-0.04 - 0.16	0.231	-0.14	-0.21 - -0.06	<0.001	0.22	0.07 - 0.37	0.004
elo^2	0.08	0.04 - 0.11	<0.001	0.01	-0.00 - 0.03	0.099	-0.17	-0.22 - -0.13	<0.001	-0.08	-0.12 - -0.05	<0.001	-0.18	-0.24 - -0.11	<0.001
mean trial number	-0.12	-0.17 - -0.08	<0.001	0.05	0.03 - 0.07	<0.001	-0.06	-0.11 - 0.00	0.054	0.04	-0.01 - 0.08	0.103	0.13	0.05 - 0.22	0.003
Random Effects															
σ^2	0.06			0.02			0.09			0.05			0.22		
τ_{00}	0.01 monkey			0.00 monkey			0.07 monkey			0.02 monkey			0.04 monkey		
ICC	0.11			0.03			0.44			0.29			0.17		
N	4 monkey			4 monkey			4 monkey			4 monkey			4 monkey		
Observations	324			324			324			324			324		
Marginal R ² / Conditional R ²	0.533 / 0.583			0.187 / 0.212			0.432 / 0.685			0.138 / 0.388			0.427 / 0.522		

Table R4: LMM models results for PT parameters for individuals with non stable Elo score dynamic

Furthermore, it is unclear what Elo rating was used when correlating it to the decision parameters. The estimation of the decision parameter is based on 1500 trials, which we assume are completed in the span of several sessions in which the Elo rating might change. Also, Elo rating might not capture the relative rank of the animals. Would using relative rank instead of Elo ratings provide a different angle to the data? (We do not expect so, but it would be advantageous to know.)

We added to the ms Figure 2 and Figure S1 that should complete the information present in the methods related to the Elo rating computation (section 'Hierarchy of Dominance'). We believe that relative ordinal rank may not fully capture changes in social hierarchy because it provides only a snapshot of an individual's position at a given time and does not account for the dynamics of wins and losses between interactions. In contrast, the Elo rating continuously updates based on each interaction, better reflecting the evolving social hierarchy. Nevertheless, we performed the analysis using relative rank and found similar results, as depicted in Table R5.

Predictors	ρ^+			ρ^-			α^+			α^-			λ		
	Estimates	CI	p	Estimates	CI	p	Estimates	CI	p	Estimates	CI	p	Estimates	CI	p
(Intercept)	-0.04	-0.22 – 0.13	0.633	-0.53	-0.63 – -0.43	<0.001	0.15	-0.18 – 0.47	0.374	-0.46	-0.71 – -0.22	<0.001	1.32	0.92 – 1.73	<0.001
age category [juvenile]	-0.06	-0.26 – 0.14	0.575	0.16	0.05 – 0.27	0.004	-0.78	-1.12 – -0.44	<0.001	-0.09	-0.37 – 0.20	0.549	-0.58	-1.05 – -0.11	0.016
age category [subadult]	-0.06	-0.13 – 0.02	0.123	-0.01	-0.05 – 0.03	0.516	0.09	-0.04 – 0.21	0.165	0.04	-0.07 – 0.14	0.490	-0.18	-0.35 – -0.00	0.049
sex [female]	0.19	-0.07 – 0.44	0.149	0.01	-0.14 – 0.15	0.925	0.22	-0.25 – 0.70	0.359	-0.16	-0.51 – 0.19	0.379	-0.12	-0.71 – 0.46	0.677
mean rank	-0.01	-0.07 – 0.04	0.569	0.03	0.00 – 0.06	0.044	0.10	0.02 – 0.18	0.020	-0.01	-0.08 – 0.07	0.871	0.16	0.04 – 0.28	0.009
mean rank ²	0.09	0.06 – 0.11	<0.001	-0.00	-0.02 – 0.01	0.649	-0.13	-0.18 – -0.09	<0.001	-0.02	-0.06 – 0.02	0.397	-0.22	-0.28 – -0.15	<0.001
mean trial number	-0.20	-0.23 – -0.18	<0.001	0.01	-0.00 – 0.02	0.208	0.08	0.04 – 0.12	<0.001	-0.02	-0.06 – 0.01	0.164	0.30	0.25 – 0.35	<0.001
Random Effects															
σ^2	0.06			0.02			0.15			0.12			0.32		
τ_{00}	0.07 monkey			0.02 monkey			0.23 monkey			0.12 monkey			0.34 monkey		
ICC	0.53			0.56			0.60			0.52			0.52		
N	17 monkey			17 monkey			17 monkey			17 monkey			17 monkey		
Observations	873			873			873			873			873		
Marginal R ² / Conditional R ²	0.358 / 0.699			0.045 / 0.578			0.141 / 0.654			0.026 / 0.530			0.190 / 0.612		

Table R5: LMM models results for PT parameters for individuals with ordinal rank instead of Elo score

2. The authors did not say how many kinds of trials they have for the lottery task. As indicated in Fig. 1, they can have $7*2 + 7*6/2 + 7*6 = 77$ trial types, and they have to estimate 6 parameters. Even though the animals performed more than 1m decisions, simply repeating the same trial type does not guarantee the estimability of the parameters. The authors need to describe their trial types more explicitly to ensure that the parameters can be recovered from the data. Explicitly, write down all the lottery information in the methods, not only in the Figures.

Figure R6: Heatmap showing the number of trials performed by each monkey (rows) for each lottery type (columns). The color intensity represents the number of trials, with the 'Total' row indicating the aggregate count across all monkeys for each lottery type.

The figure R6 provides an overview of the sampling coverage for each animal and task. The detailed description of each lottery type has been presented in our previous publications (Nioche et al., 2019; 2021). For clarity, we also summarize it here:

- Type 1 - Loss vs. Gain ($x_1 > 0, x_2 < 0, p_1 = p_2$): 36 pairs
- Type 2 - Gain discrimination ($p_1 = p_2, x_1 > x_2 > 0$): 12 pairs
- Type 3 - Loss discrimination ($p_1 = p_2, x_1 < x_2 < 0$): 12 pairs
- Type 4 - Gain probability ($p_1 > p_2, x_1 = x_2 > 0$): 12 pairs
- Type 5 - Loss probability ($p_1 < p_2, x_1 = x_2 < 0$): 18 pairs
- Type 6 - Gain, no stochastic dominance ($p_1 < p_2, x_1 > x_2 > 0$): 18 pairs
- Type 7 - Loss, no stochastic dominance ($p_1 < p_2, x_1 < x_2 < 0$): 18 pairs

Lottery Types 1-5 correspond to conditions with stochastic dominance, while Types 6 and 7 are non-dominant lotteries, used to assess decision-making under risk.

In addition, in the Methods section now entitled '*Task and Estimation of the Prospect Theory (PT) parameters*', we added the following information:

"The task includes different types of lotteries designed to test three conditions (Fig. S2), both in the gain and loss domains: same-probability/different-outcome (quantity discrimination), same-

outcome/different-probability (probability discrimination), and trade-offs between probability and outcome magnitude (risk-attitude assessment)."

Furthermore, it will be advantageous to show that Tonkean macaques also show first-order stochastic dominance. That is, while holding reward probability or magnitude constant across choices, the animals chose the dominating option more often than chance.

We thank the reviewer for this feedback, we added Figure S2 to the revised ms to illustrate this point. Monkeys demonstrated sensitivity to the difference in expected value (EV) between two options under various conditions, as shown in the following figure. This sensitivity was evident when probabilities were equal but amounts differed (first column), when amounts were equal but probabilities differed (second column), and in scenarios involving a trade-off between quantity and probability (third column). This pattern held true in both gain and loss domains. The figure illustrates that animals chose the dominant option significantly more often than chance, reflecting adaptive decision-making based on EV differences.

Figure S2: Monkeys' decision-making across six lottery choice tasks. Each subplot examines consideration of expected value (EV) differences under different conditions: (task id #2 #3) when probabilities are equal but amounts differ, (task id #4 #5) when amounts are equal but probabilities differ, and (task id #6 #7) when there is a trade-off between quantity and probability. Blue lines (top row) represent choices involving gains, orange lines (bottom row) represent choices involving losses. Scattered points show individual monkey binary choices

(0/1) plotted against EV differences. Thin lines represent individual sigmoid fits (one line per monkey), while thick colored lines show the mean function across all monkeys. The panels in the first two columns show probability of choosing the option with higher expected value, while panels in the last column show probability of choosing the riskiest option.

3. Concerning the LMM model, the authors did not say why they chose this form of model. Such as, Why is only the elo-rating included as a second-order term? Why did the random effect only include a random intercept but not a random slope? The authors need to justify the model specification, perhaps by performing model selection.

We thank the reviewer for raising this point, the revised ms justified better our model construction. Concerning the LMM model, factors were selected based on their hypothesized potential impact on risk attitude. Our working hypothesis is that the Elo-rating includes a second-order term in order to capture the nonlinear effects of social rank that has been shown on health and various macaque behaviors, including boldness (see McCowan *et al.*, 2022). On the other hand, other biological parameter such as age and sex should not display such nonlinear effect.

Regarding random factors, we choose to include random intercept to take into account the nested nature of our dataset (all points are not independent as individuals performed thousands of trials). Including also a random slope in the models would not be relevant to our analysis as it would assume that we want to model U-shape relationships at individual level. This cannot be achieved as an individual does not experience the full range of Elo-rating in their lifetime (see answer to Reviewer #2 point (6) for more details).

We added the following paragraph in the Material and Methods (in the LMM modelling subsection):

“A random intercept was included to account for the non-independence of data points, as each individual completed thousands of trials. Random slope was not included, as this would imply that all individuals respond similarly across the predictor, whereas in reality each individual was exposed to only a limited range of social ranks during data collection.”

Minor comments

1. Concerning the exclusion of animals, the authors excluded six animals from the raw data. They said the excluded animals have a severe side bias. We wonder if it is possible to incorporate this side bias into their model, and aid in the model selection process. This analysis can be complemented by showing that despite side biases (regardless of their magnitude), the animals chose the dominating option.

Side bias is already incorporated in the PT models through the x_0 parameter, which also facilitates the estimation of other PT parameters, such as risk aversion. Using the estimated parameters, we applied the interquartile range (IQR) method to exclude trial periods where PT parameters were extreme outliers. Individuals excluded by this method were those exhibiting

a strong side bias and, consequently, failing to choose the dominating option in control trials (See *Data Filtering* section in the Methods for more details).

2. In the label of Fig. 1:

a. We think it should read ...a 25% chance ... in the sentence “If the monkey chooses the left pie chart, it will have half a chance of getting 1 token. If it chooses the right pie chart, it will have a 75% chance to get 2 tokens.”

b. What is half a chance? Please revise.

3. In Figures 2 and 3, place the label, such as risk aversion, risk seeking, etc, near the axis.

We thank the reviewer for picking this up, this has been corrected.

4. It is unclear how using the MALT, which ‘track(s) thousands of decisions over several years’, enables daily updates of dominance hierarchies.

Each day that individuals use the MALT allows us to observe conflicts between them over access, which are then used to calculate an Elo score reflecting the group’s social hierarchy (see Ballesta *et al.*, 2021 and response to Reviewer #2).

5. It’s unclear how “Adopting the PT framework, we analyzed the decisions of 18 Tonkean macaques across 1,380,190 trials” allows the longitudinal study of risky decision making. Longitudinally would come from testing over time, but the sentence refers to the number of trials.

We apologize for the confusion. Information about the periods of data collection have been added in the result and methods section. The following sentences were added:

“Adopting the PT framework, we analyzed the decisions of 18 Tonkean macaques across 1,380,190 trials **over a three-year period from February 2020 to June 2023**, allowing us to study risk attitudes longitudinally (**Fig. 1**)”

“The group included 28 individuals, data from 24 individuals (13 males) were considered **over a three-year period from February 2020 to June 2023**”

Reviewer #4 (Remarks to the Author):

I co-reviewed this manuscript with one of the reviewers who provided the listed reports. This is part of the Communications Biology initiative to facilitate training in peer review and to provide appropriate recognition for Early Career Researchers who co-review manuscripts.

Thank you for acknowledging us.